# Angular Regularization for Positive-Unlabeled Learning on the Hypersphere

**Vasileios Sevetlidis***  *vasiseve@athenarc.gr*
*Athena RC*
*Institute for Language and Signal Processing*
*University Campus Kimmeria*
*Xanthi, GR67100, Greece*

*Democritus University of Thrace*
*Dept. Production and Management Engineering*
*Vas. Sofias 17*
*Xanthi, GR67100, Greece*
*\* Corresponding Author*

**George Pavlidis**  *gpavlid@athenarc.gr*
*Athena RC*
*Institute for Language and Signal Processing*
*University Campus Kimmeria*
*Xanthi, GR67100, Greece*

**Antonios Gasteratos**  *agaster@pme.duth.gr*
*Democritus University of Thrace*
*Dept. Production and Management Engineering*
*Vas. Sofias 17*
*Xanthi, GR67100, Greece*

**Reviewed on OpenReview:** *https://openreview.net/forum?id=XQhOOLy6el*

## Abstract

Positive–Unlabeled (PU) learning addresses classification problems where only a subset of positive examples is labeled and the remaining data is unlabeled, making explicit negative supervision unavailable. Existing PU methods often rely on negative-risk estimation or pseudo-labeling, which either require strong distributional assumptions or can collapse in high-dimensional settings. We propose AngularPU, a novel PU framework that operates on the unit hypersphere using cosine similarity and angular margin. In our formulation, the positive class is represented by a learnable prototype vector, and classification reduces to thresholding the cosine similarity between an embedding and this prototype—eliminating the need for explicit negative modeling. To counteract the tendency of unlabeled embeddings to cluster near the positive prototype, we introduce an angular regularizer that encourages dispersion of the unlabeled set over the hypersphere, improving separation. We provide theoretical guarantees on the Bayes-optimality of the angular decision rule, consistency of the learned prototype, and the effect of the regularizer on the unlabeled distribution. Experiments on benchmark datasets demonstrate that AngularPU achieves competitive or superior performance compared to state-of-the-art PU methods, particularly in settings with scarce positives and high-dimensional embeddings, while offering geometric interpretability and scalability.

# 1 Introduction

Positive–Unlabeled (PU) learning addresses classification scenarios in which labeled negatives are scarce or infeasible to obtain (Bekker & Davis, 2020; Elkan & Noto, 2008). Such settings are common in domains like medical diagnosis, sentiment analysis, and anomaly detection, where negative examples can be costly, ambiguous, or ethically constrained. Formally, we are given a set of labeled positive instances and a large pool of unlabeled data containing an unknown mix of positives and negatives; the task is to learn a classifier that separates the two without explicit negative supervision.

Existing PU methods typically rely on *surrogate modeling* of the negative class(Yu et al., 2002; Liu et al., 2003; Hsieh et al., 2019; Gong et al., 2021; Yuan et al., 2025). Risk-based approaches (e.g., unbiased or non-negative PU risk estimators and their imbalanced variants) decompose the classification risk into terms involving only positives and unlabeled data, using an assumed class prior to correct for the implicit treatment of all unlabeled examples as negatives (Du Plessis et al., 2014; Christoffel et al., 2016; Ramaswamy et al., 2016; Kiryo et al., 2017). While theoretically sound, these methods are sensitive to prior estimation errors and degrade when the learned representation is insufficiently discriminative. EM-style and prototype-contrastive methods, in contrast, generate pseudo-labels for the unlabeled set and alternate between representation learning and classifier updates; however, such alternating schemes are prone to confirmation bias—early mislabeling of negatives as positives can cause cascading errors—and often require auxiliary mechanisms such as momentum queues, hard-negative mining, or contrastive pair construction(Arazo et al., 2020; Cascante-Bonilla et al., 2021); contrastive mechanics: (He et al., 2020; Wu et al., 2018; Chen et al., 2020b). Boundary-or anomaly-based methods take a different route by synthesizing negatives or convexifying the positive set in latent space, but these heuristics can fail when the positive manifold is complex or non-convex (Schölkopf et al., 2001; Ruff et al., 2018).

A common limitation across these approaches is the reliance on either explicit negative modeling or iterative pseudo-negative construction, both of which can be unstable in high-dimensional feature spaces. Moreover, most methods treat representation learning as secondary, leaving the geometric structure of the embedding space underexploited.

We take a *geometry-first* approach: we explicitly model only the positive class and treat all other instances as dispersed over the hypersphere. Concretely, we propose a neural encoder that maps inputs to the unit hypersphere and represents the positive class using a directional scoring function based on cosine similarity to a learnable prototype vector. This yields a simple *angular score*

$$s(z) = \kappa \, \mu^\top z,$$

which provides a geometrically meaningful decision rule and eliminates the need for explicit negative modeling, pseudo-negatives, or class-prior estimation.

To reduce false positive clustering near the positive prototype, we introduce an *unlabeled-only angular regularizer* that encourages embeddings of unlabeled examples to be well-dispersed over the hypersphere. The encoder and prototype are learned jointly in a single-stage, end-to-end optimization.

Our contributions are:

1. **Hyperspherical PU modeling:** We formulate PU learning in a hyperspherical embedding space, using cosine similarity to a learned positive prototype as the classification score.

2. **Unlabeled-only angular regularization:** We regularize only the unlabeled set toward hyperspherical uniformity, mitigating false positive clustering and improving separation in high-dimensional spaces.

3. **Simplicity and stability:** Our method is a single-stage, geometry-driven approach that avoids momentum queues, contrastive pairs, pseudo-negative heuristics, or class-prior estimation, while remaining theoretically grounded and scalable.

Experiments on four benchmark datasets (CIFAR-10, STL-10, SVHN, and ADNI) show that our method achieves competitive or superior performance compared to state-of-the-art PU baselines, particularly in

regimes with few labeled positives, while offering a stable and interpretable geometric framework for PU learning.

## 2  Related Work

PU learning has been studied extensively over the past two decades, with methods broadly falling into three categories: *risk-based approaches*, *representation-learning approaches*, and *boundary/anomaly-based methods*. Below, we summarize the most relevant works and position our method within this landscape.

Unbiased risk estimation (uPU) (Du Plessis et al., 2014) and its non-negative variant nnPU (Kiryo et al., 2017) form the basis of many PU approaches, decomposing the classification risk into terms involving only positives and unlabeled data. While theoretically grounded, these estimators are sensitive to class prior estimation, can overestimate negative risk, and may overfit when representation quality is low. Imbalanced nnPU (Su et al., 2021) addresses class imbalance via reweighting, effectively oversampling positives in PN settings. However, it still assumes accurate knowledge of the class prior $\pi_1$, inherits nnPU's non-negativity heuristic, and does not directly improve the geometry of the learned embeddings—a crucial factor in high-dimensional settings. In contrast, our method avoids negative-risk estimation and reweighting altogether, directly modeling the positive distribution on the hypersphere and regularizing the unlabeled set toward angular uniformity.

To overcome the representation bottleneck of purely risk-based methods, recent works integrate metric learning or contrastive objectives into PU frameworks. WConPU (Yuan et al., 2025) combines weighted contrastive learning with prototype-based classification, alternating between prototype updates, pseudo-labeling, and classifier training, while also performing hard-negative mining from a momentum queue. This improves embeddings but introduces multiple interdependent components and requires a class-prior estimate. Our approach shares the goal of compact positive clustering but replaces the pseudo-negative pipeline with a probabilistic vMF prototype learned end-to-end, requiring neither EM-style alternation nor memory queues.

In the broader weakly supervised learning literature, PiCO (Wang et al., 2022) addresses partial-label learning via label disambiguation and contrastive representation learning. Subsequent work has shown that contrastive learning often produces mixtures of vMF distributions in embedding space. However, PiCO assumes candidate label sets for each sample and is not directly applicable to the PU setting (Yuan et al., 2025). Our work makes the vMF modeling explicit for positives in PU, eliminating the need for candidate labels, contrastive pair construction, or label disambiguation.

Ensemble-diversity self-training (Odonnat et al., 2024) encourages disagreement among hypotheses on unlabeled data to mitigate selection bias. Our approach is complementary: we enforce dispersion in representation space for unlabeled data while aligning positives to a spherical prototype. The alignment–uniformity lens of (Wang & Isola, 2020) directly motivates our design: positive alignment to $\mu$ and unlabeled uniformity on $\mathbb{S}^{d-1}$.

Another direction infers negatives by modeling the positive region and treating outliers as negative. Dense-PU (Sevetlidis et al., 2024) trains a convolutional autoencoder on positives, interpolates in the latent space, and uses convex hull or dense region boundaries to identify negative candidates for downstream PN training. This two-stage approach depends on SCAR assumptions and convexity heuristics, which may fail for complex, non-convex manifolds. In contrast, our method is end-to-end, avoids explicit negative generation, and uses a principled probabilistic model for the positive class on the hypersphere.

Most existing PU methods either (i) assume a specific distribution for the negative class, (ii) iteratively construct pseudo-negatives, or (iii) decouple representation learning from classifier optimization. Each of these choices brings well-known drawbacks: risk-based estimators are sensitive to class-prior misspecification and do little to shape the embedding space; pseudo-labeling pipelines amplify early mistakes through confirmation bias; and contrastive methods require auxiliary machinery such as momentum queues and apply uniformity constraints indiscriminately, often eroding positive compactness. Our geometry-first framework sidesteps all three pitfalls by modeling only the positive class as a von Mises–Fisher distribution on the hypersphere and enforcing uniformity exclusively on the unlabeled set. While our method does use a constant soft label of 0.5 for unlabeled data—reflecting maximum uncertainty—it avoids dynamic or EM-style

Table 1: Comparison of representative PU learning methods. "Prior" indicates reliance on a class-prior estimate; "Neg. model" means an explicit distributional assumption for the negative class; "Pseudo-neg." indicates iterative pseudo-negative labeling; "Contrastive" denotes explicit contrastive pair construction; "vMF$^+$" means an explicit von Mises–Fisher model for positives; "U-uniform" indicates an unlabeled-only uniformity regularizer; "End-to-end" denotes single-stage training without multi-phase pipelines.

| Method | Prior | Neg. model | Pseudo-neg. | Contrastive | vMF$^+$ | U-uniform | End-to-end |
|---|---|---|---|---|---|---|---|
| nnPU (Kiryo et al., 2017) | ✓ | ✗ | ✗ | ✗ | ✗ | ✗ | ✓ |
| Imbalanced nnPU (Su et al., 2021) | ✓ | ✗ | ✗ | ✗ | ✗ | ✗ | ✓ |
| WConPU ()yuan2025weighted | ✓ | ✗ | ✓ | ✓ | ✗ | ✗ | ✗ |
| PiCO Wang et al. (2022) | ✗ | ✗ | ✓ | ✓ | ✗ | ✗ | ✗ |
| Dense-PU (Sevetlidis et al., 2024) | ✗ | ✗ | ✓ | ✗ | ✗ | ✗ | ✗ |
| **Ours** | ✗ | ✗ | ✗[1] | ✗ | ✓ | ✓ | ✓ |

pseudo-labeling and relies solely on a probabilistic directional score for classification. The result is a simple, stable, and theoretically grounded PU learner that aligns its inductive bias directly with the geometry of high-dimensional embeddings.

# 3  Proposed Method

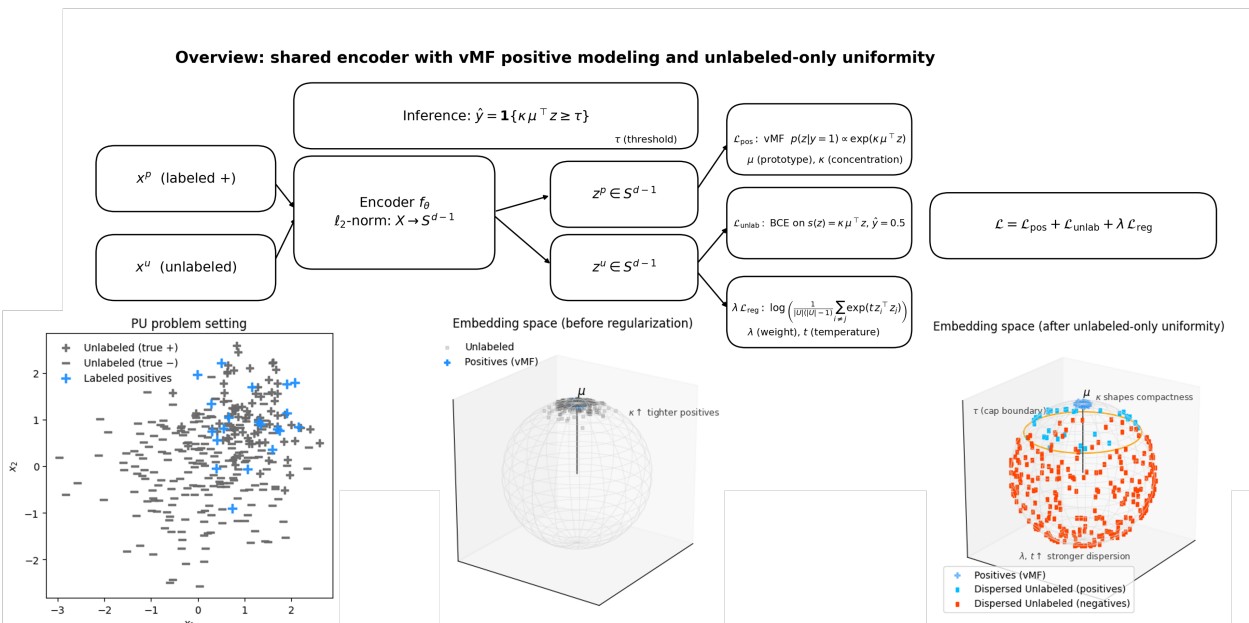

Figure 1: Overview of our hyperspherical PU method. A shared encoder $f_\theta$ maps labeled positives $x^p$ and unlabeled samples $x^u$ onto the unit hypersphere. Positives are pulled toward a learnable direction $\mu$ via a cosine-based alignment loss ($\mathcal{L}_{\text{pos}}$). Unlabeled samples are trained with a symmetric BCE loss ($\mathcal{L}_{\text{unlab}}$) and dispersed via an angular uniformity regularizer ($\mathcal{L}_{\text{reg}}$). Right: regularization increases angular separation, yielding compact positives and spread-out unlabeled embeddings.

We address the PU learning problem, where only a small set of labeled positives $\mathcal{D}_p$ and a large set of unlabeled samples $\mathcal{D}_u$ are available during training. The unlabeled set contains an unknown mixture of positives and negatives, with no access to explicitly labeled negatives. This setup commonly arises in settings where negative categories are heterogeneous, ill-defined, or impractical to label exhaustively.

---

[1]We use a fixed uncertainty label of 0.5 for the unlabeled set, rather than dynamic pseudo-labeling or negative mining.

Our method learns a neural encoder $f_\theta : \mathcal{X} \to \mathbb{S}^{d-1}$ that maps inputs to $\ell_2$-normalized feature vectors on the unit hypersphere (Mardia & Jupp, 2009; Dhillon & Sra, 2003) (see Figure 1). This geometry is well-suited for angular decision rules based on cosine similarity. Rather than modeling a full density, we score each embedded vector $z = f_\theta(x)$ via its similarity to a learnable direction $\mu \in \mathbb{S}^{d-1}$, using a scaled cosine similarity. This directional score draws inspiration from the von Mises–Fisher distribution, which defines a density over the hypersphere with normalization constant $C_d(\kappa)$ (see Appendix A for details). While we do not require this constant in training, it is used in the log-likelihood derivation and analytical bounds:

$$s(z) = \kappa \, \mu^\top z,$$

where $\kappa > 0$ is a fixed scaling factor controlling the sharpness of the score. This directional score serves as a soft classifier, with $\mu$ acting as a prototype for the positive class, which is updated by gradient descent and re-projected onto the unit sphere after each optimization step to enforce $\|\mu\| = 1$. That is, we perform:

$$\mu \leftarrow \frac{\mu}{\|\mu\|_2} \quad \text{after each update.}$$

We adopt a geometric inductive bias that encourages unlabeled samples to be dispersed over the sphere, inspired by the idea that negatives occupy diverse regions of embedding space while positives tend to cluster near $\mu$. This formulation is consistent with a vMF–uniform generative model. In such a setting, the Bayes-optimal decision rule reduces to thresholding the inner product $\mu^\top z$. While this motivates our directional scoring formulation, in practice we use empirical threshold selection without requiring class priors or normalization constants.

The model is trained by minimizing a loss composed of three terms. For labeled positives $P \subset \mathcal{D}_p$, we encourage alignment with $\mu$:

$$\mathcal{L}_{\text{pos}} = -\frac{1}{|P|} \sum_{i \in P} \kappa \, \mu^\top z_i. \tag{1}$$

For unlabeled samples $U \subset \mathcal{D}_u$, we apply a symmetric binary cross-entropy loss with neutral supervision, reflecting maximum uncertainty (i.e., $\mathbb{E}[y] = 0.5$):

$$\mathcal{L}_{\text{unlab}} = -\frac{1}{|U|} \sum_{j \in U} \left[ 0.5 \log \sigma(\ell_j) + 0.5 \log(1 - \sigma(\ell_j)) \right], \tag{2}$$

where $\ell_j = \kappa \mu^\top z_j$ and $\sigma$ denotes the sigmoid function. This uncertainty-driven term avoids overconfident updates on ambiguous unlabeled points.

To mitigate false positives clustering near the prototype, we regularize the unlabeled set via an angular dispersion term:

$$\mathcal{L}_{\text{reg}} = \log \left( \frac{1}{|U|(|U| - 1)} \sum_{i \neq j} e^{t \, z_i^\top z_j} \right), \tag{3}$$

where $t > 0$ is a temperature hyperparameter. This regularizer encourages decorrelation among unlabeled embeddings, improving separability by promoting diversity in feature space.

The final loss combines the three components:

$$\mathcal{L} = \mathcal{L}_{\text{pos}} + \mathcal{L}_{\text{unlab}} + \lambda \, \mathcal{L}_{\text{reg}}, \tag{4}$$

where $\lambda$ controls the strength of regularization. The encoder $f_\theta$ and prototype $\mu$ are learned jointly via back-propagation. $\mu$ is initialized randomly on $\mathbb{S}^{d-1}$ and renormalized after each update. The scaling parameter $\kappa$ is fixed throughout training and treated as a hyperparameter.

To improve robustness to uncertain unlabeled samples, we incorporate a soft weighting scheme based on a fixed or learnable *angular margin* $m \in [-1, 1]$, which acts as a similarity threshold. The idea is to assign

greater influence to unlabeled embeddings that lie closer to the positive prototype $\mu$ in angular space. Specifically, for each $z \in U$, we define a soft weight:

$$w(z) = \sigma\left(\alpha \cdot (\mu^\top z - m)\right), \tag{5}$$

where $\sigma$ is the sigmoid function and $\alpha > 0$ controls the sharpness of the transition. Samples with $\mu^\top z \gg m$ (i.e., close to the prototype) receive higher weights.

We then modify the unlabeled loss to incorporate these weights[2]:

$$\mathcal{L}_{\text{unlab}} = -\frac{1}{|U|} \sum_{j \in U} w(z_j) \cdot \left[0.5 \log \sigma(\ell_j) + 0.5 \log(1 - \sigma(\ell_j))\right]. \tag{6}$$

Because the unlabeled loss gradient satisfies

$$\frac{\partial \mathcal{L}_{\text{unlab}}}{\partial \ell} = w(z)\left[\sigma(\ell) - 0.5\right],$$

the weighting function $w(z)$ increases the tendency to neutralize overly confident, high-similarity unlabeled points, thereby counteracting false-positive collapse near $\mu$.

At test time, a score $s(z) = \kappa\, \mu^\top z$ is computed for each sample. The decision threshold $\tau$ is selected via F1 maximization on a held-out validation set, and predictions are made by:

$$\hat{y} = \begin{cases} 1 & \text{if } s(z) \geq \tau \\ 0 & \text{otherwise} \end{cases}.$$

## 4 Theoretical Justification

Our theoretical analysis serves three purposes: (i) to motivate our directional scoring function based on a generative vMF–uniform model, (ii) to justify the learnability of the positive prototype $\mu$ under realistic assumptions, and (iii) to interpret our cosine regularizer as encouraging dispersion of the unlabeled set. While our implementation does not rely on priors or density estimation, these results illuminate the inductive biases built into our model.

### 4.1 Directional decision boundary under the vMF–uniform model

We consider a generative setting where positives follow a von Mises–Fisher (vMF) distribution on the hypersphere and negatives are uniform (Mardia & Jupp, 2009; Dhillon & Sra, 2003; Sra, 2016).

**Proposition 4.1** (MAP rule for vMF–uniform). *Let $z \in \mathbb{S}^{d-1}$, with $p(z \mid Y{=}1) = C_d(\kappa)\exp(\kappa\,\mu^\top z)$ and $p(z \mid Y{=}0) = U_d$, and prior $\pi = \Pr(Y{=}1)$. Then the Bayes–optimal classifier is a threshold on the inner product:*

$$\mu^\top z \;\geq\; \tau \;:=\; -\frac{1}{\kappa}\left(\log C_d(\kappa) - \log U_d + \log \frac{\pi}{1-\pi}\right).$$

*Proof sketch.* Bayes' rule gives

$$\log \frac{p(Y{=}1 \mid z)}{p(Y{=}0 \mid z)} = \kappa\,\mu^\top z + \log C_d(\kappa) - \log U_d + \log \frac{\pi}{1-\pi}.$$

The MAP decision sets this $\geq 0$, yielding the stated threshold. Full constants/derivation are in Appx. A.

vMF level sets are spherical caps centered at $\mu$; the MAP acceptance region is the cap $\{\mu^\top z \geq \tau\}$.

---

[2]In ablations, we test both fixed-margin settings (e.g., $m = 0.5$) and learnable-margin variants, where $m$ is optimized during training and constrained to the interval $[-1, 1]$. The adaptive weights guide learning by prioritizing unlabeled instances closer to the decision boundary, reducing the impact of noisy or uninformative examples.

**Remark 1** (Isotropic negatives). *If $p(z \mid Y{=}0)$ is rotation-invariant on $\mathbb{S}^{d-1}$, then $\log \frac{p(z|1)}{p(z|0)} = \kappa \mu^\top z + c$ for a constant $c$, so the Bayes rule is still a threshold on $\mu^\top z$.*

Our score is $s(z) = \alpha(\mu^\top z - m)$, so thresholding $s$ is equivalent to thresholding $\mu^\top z$ with $s(z) \geq \tau_s$ and $\tau_s = \alpha(\tau - m)$. In practice we estimate $\tau_s$ using a PU-valid calibration on unlabeled scores (mixture-proportion based), not supervised F1. See §5 and Appx. D.

## 4.2 Learnability of the prototype

We now show that the directional prototype $\mu$ can be reliably estimated from labeled positives alone, under the assumption that $\kappa$ is fixed.

**Lemma 4.1** (Consistency of the Positive Prototype). *Let $z_1, \ldots, z_n \sim vMF(\mu, \kappa)$, and let $\bar{r}_n = \frac{1}{n}\sum_{i=1}^{n} z_i$ be the sample mean. Then the maximum likelihood estimate of $\mu$ is $\hat{\mu}_n = \bar{r}_n/\|\bar{r}_n\|$, and:*

$$\hat{\mu}_n \xrightarrow{p} \mu \quad as\ n \to \infty.$$

This shows that, under fixed $\kappa$ and fixed encoder $f_\theta$, the prototype $\mu$ converges in probability to the true mean direction (Mardia & Jupp, 2009). In practice, we jointly optimize $\mu$ and $f_\theta$, so this consistency result holds approximately and serves as a justification for prototype stability when the positive distribution is concentrated.

*Proof.* The MLE of the vMF mean direction is known to converge to the population mean direction (Mardia & Jupp, 2009). See Appendix B for derivation.

## 4.3 Interpretation of the cosine uniformity regularizer

The previous results explain why scoring via $\mu^\top z$ can separate positives and negatives—if $\mu$ is well estimated and negatives are not concentrated around $\mu$. However, in practice, the unlabeled set may contain false positives that cluster near the prototype, especially early in training.

To reduce this risk, we include a regularization term that promotes angular dispersion of the unlabeled set. While it is inspired by hyperspherical uniformity, we do not assume the unlabeled set is actually uniform.

**Proposition 4.2** (Dispersion via Cosine Regularization). *Let $U = \{z_1, \ldots, z_n\}$ be embeddings of unlabeled samples on $\mathbb{S}^{d-1}$. Then the regularizer*

$$\mathcal{L}_{reg} = \log\left(\frac{1}{n(n-1)} \sum_{i \neq j} e^{t z_i^\top z_j}\right)$$

*is minimized when the pairwise cosine similarities $z_i^\top z_j$ are low on average, encouraging angular spread.*

Minimizing $\mathcal{L}_{\text{reg}}$ reduces high pairwise cosine similarities among unlabeled embeddings—i.e., it promotes angular dispersion (soft repulsion)—without assuming exact hyperspherical uniformity. While it does not enforce true uniformity, it reduces redundancy in the latent space and improves separation.

*Proof.* Since $z_i^\top z_j \in [-1, 1]$, and $e^{t z_i^\top z_j}$ increases with similarity, the sum is minimized when the $z_i$ are dispersed (i.e., pairwise similarities are small). See Appendix C for further analysis.

## 4.4 Regularization scaling and bound

We provide a loose upper bound on the regularizer to ensure it does not dominate the overall objective.

**Corollary 4.1** (Upper Bound on Regularization Term). *Let $z_i \in \mathbb{S}^{d-1}$ for $i = 1, \ldots, n$. Then:*

$$\mathcal{L}_{reg} \leq \log\left(e^t\right) = t.$$

This holds in the extreme case where all cosine similarities are maximal, i.e., $z_i^\top z_j = 1$. In practice, $\mathcal{L}_{\text{reg}}$ remains well-behaved and is scaled by a tunable factor $\lambda$.

*Proof.* The maximum value of each term $e^{t\, z_i^\top z_j}$ is $e^t$. There are $n(n-1)$ such terms, so the average is at most $e^t$. Taking log gives $t$.

## 5 Experimental Evaluation

We evaluate the proposed vMF PU learning framework on a range of benchmark datasets: CIFAR-10, STL-10, SVHN, and ADNI. These datasets span natural images, digit recognition, and medical imaging, allowing us to assess performance across varying input complexity and domain characteristics. The PU setting is simulated by providing a small labeled positive set and treating the remainder of the data as unlabeled, containing a mixture of positives and negatives.

### 5.1 Experimental Protocol

To ensure comparability with prior work, we replicate the experimental setup of Yuan et al. (2025), including dataset partitions, positive/unlabeled (PU) label ratios, and evaluation metrics. This enables direct comparisons with established PU learning baselines.

Each input is encoded onto the unit hypersphere $\mathbb{S}^{d-1}$ via a trainable encoder, as described in Section 3. Training optimizes a von Mises-Fisher-based loss for labeled positives, binary cross-entropy for unlabeled data, and—when applicable—the cosine uniformity regularizer. We report the following evaluation metrics on the test set: F1 score, precision, recall, accuracy, area under the ROC curve (AUC), and average precision (AP), consistent with prior literature.

Unless otherwise stated, all experiments use a validation split (10% of the training data) to tune the decision threshold for binary classification by maximizing F1 score.

### 5.2 Datasets

**CIFAR-10** consists of 60,000 color images (32×32) across 10 object classes (Krizhevsky et al., 2009). Following prior work, we group *airplane*, *automobile*, *ship*, and *truck* as the positive (vehicle) class, and the remaining classes as negative (animals). We sample 1,000 positive examples to serve as labeled data; the rest of the training set is used as unlabeled data containing a mixture of positives and negatives. We adopt the standard 50,000/10,000 train/test split.

**STL-10** contains 13,000 labeled images (96×96) across 10 classes (Coates et al., 2011). We group *airplane*, *car*, and *truck* as positives, and all other classes as negatives. A total of 500 positive examples are randomly selected as labeled data, with the rest used as unlabeled. Due to its higher resolution and more complex scenes, STL-10 poses a greater challenge compared to CIFAR-10.

**SVHN** comprises over 600,000 real-world digit images from street views (Netzer et al., 2011). Even digits (0, 2, 4, 6, 8) are treated as positives and odd digits as negatives. We randomly sample 1,000 even digits as labeled positives, while treating the remainder of the training set as unlabeled. The test set includes 26,032 labeled images.

**ADNI (Alzheimer's Spectrum).** The Alzheimer's Disease Neuroimaging Initiative (ADNI) provides structural MRI scans of healthy controls (NC) and subjects across the Alzheimer's disease spectrum (AD, MCI, etc.)(Jack Jr et al., 2008; Petersen et al., 2010). We treat all NC spectrum cases as positives and the rest as negatives. From the positive class, 768 scans are randomly selected as labeled data. The remainder of the training set (including both NC and unlabeled AD spectrum samples) forms the unlabeled pool. Evaluation is conducted on a held-out test set with ground-truth diagnostic labels.

### 5.3 Training Details

For image datasets (CIFAR-10, STL-10, SVHN), we adopt a VGG11-BN encoder (Simonyan & Zisserman, 2014) pretrained on ImageNet (Deng et al., 2009). For the ADNI dataset (Jack Jr et al., 2008; Petersen et al., 2010), we use a two-layer multilayer perceptron (MLP) with ReLU activations. All models are trained for 15 epochs using the Adam optimizer with a learning rate of $10^{-4}$ and a batch size of 128. The encoder maps inputs to a $d$-dimensional hypersphere, with $d = 128$ unless stated otherwise. We apply dropout with a rate of 0.2 to the final embedding layer. The cosine uniformity regularizer is scaled by temperature $t = 2.0$. The concentration parameter $\kappa$ of the von Mises-Fisher distribution is tuned per dataset as a hyperparameter. Unless otherwise specified, all components of the encoder are updated end-to-end, including learnable margins where applicable.

### 5.4 Baselines

We compare against a representative set of PU learning approaches, including EM-based methods, importance-weighted PU learning, and surrogate loss methods, using results directly from Yuan et al. (2025) where applicable. This ensures identical data splits and evaluation criteria across all methods:

- **uPU** (Du Plessis et al., 2014): Risk-based PU method using an unbiased risk estimator under the SCAR assumption, with theoretical analysis showing that convex losses can bias the boundary, while non-convex losses (e.g., ramp) avoid this bias.

- **nnPU** (Kiryo et al., 2017): Extends uPU by enforcing non-negativity of the empirical risk to prevent overfitting in flexible models, enabling stable training of deep networks with PU data.

- **Rank Pruning** (Northcutt et al., 2017): Estimates label noise rates and prunes mislabeled examples using high-confidence samples ranked by predicted probability, improving robustness in both PU and noisy-label settings.

- **PUSB** (Kato et al., 2019): PU learning under selection bias, relaxing the SCAR assumption via the "invariance of order" property and density ratio estimation, with thresholding for final classification.

- **puNCE** (Acharya et al., 2022): Adversarial PU framework where a discriminator separates labeled positives from unlabeled data, and the generator learns embeddings to fool the discriminator, aided by label distribution estimation.

- **Self-PU** (Chen et al., 2020c): Self-paced PU framework that progressively labels confident positives and negatives, reweights uncertain samples via meta-learning, and applies self-distillation to enforce consistency.

- **VPU** (Chen et al., 2020a): Variational PU method that measures divergence between the positive distribution and model predictions without estimating the class prior, regularized via Mixup-based consistency.

- **Dist-PU** (Zhao et al., 2022): Aligns predicted and ground-truth label distributions, with entropy minimization and Mixup to reduce confirmation bias and avoid degenerate solutions.

- **PiCO** (Wang et al., 2022): Contrastive label disambiguation method combining representation learning with prototype-based pseudo-label refinement in an alternating optimization scheme.

- **Dense-PU** (Sevetlidis et al., 2024): Density-based negative mining in latent space to reduce false positives, by iteratively identifying dense negative clusters from unlabeled data.

- **ImbPU** (Su et al., 2021): Prototype-based PU approach estimating positive and negative centroids and refining pseudo-labels via local neighborhood consistency.

- **aPU** (Hammoudeh & Lowd, 2020): Proposes a two-step method using surrogate negatives and a recursive risk estimator for learning from positive and unlabeled data even when the positive data is arbitrarily biased, by assuming the negative class distribution remains fixed.

- **PUbN** (Hsieh et al., 2019): PU learning with biased negative data, introducing a correction term to adjust the risk estimator when negative samples come from a biased distribution.

- **WconPU** (Yuan et al., 2025): Distribution-weighted PU learning balancing the positive and unlabeled risks via dynamically learned weights, improving robustness under varying class priors.

## 5.5 Results

Table 2: Performance of the proposed method vs. baseline methods. Results are reported for F1 Score, Accuracy, Precision, Recall, AUC, and AP (the best score is marked in **bold**).

| | Method | Accuracy | Precision | Recall | F1 | AUC | AP |
|---|---|---|---|---|---|---|---|
| *CIFAR-10* | uPU | 88.41±0.41 | 87.21±2.09 | 83.02±1.98 | 85.12±0.43 | 94.98±0.62 | 92.71±1.08 |
| | nnPU | 88.91±0.43 | 86.21±1.02 | 86.03±1.22 | 86.11±0.49 | 95.13±0.55 | 92.51±1.32 |
| | RP | 88.74±0.16 | 86.02±1.05 | 85.72±1.61 | 85.93±0.30 | 95.21±0.23 | 93.01±0.61 |
| | PUSB | 88.97±0.39 | 86.15±0.56 | 86.22±0.45 | 86.18±0.51 | 95.15±0.50 | 92.44±1.34 |
| | PUbN | 89.83±0.30 | 87.85±0.98 | 86.56±1.87 | 87.18±0.54 | 94.44±0.35 | 91.28±1.11 |
| | Self-PU | 89.31±0.56 | 86.26±0.76 | 87.22±2.16 | 86.77±1.12 | 95.52±0.46 | 93.31±1.02 |
| | aPU | 89.09±0.44 | 86.31±1.33 | 86.33±0.71 | 86.41±0.41 | 95.11±0.39 | 92.42±1.22 |
| | VPU | 87.89±0.56 | 86.71±1.46 | 82.88±2.93 | 84.42±1.12 | 94.55±0.46 | 92.02±0.69 |
| | ImbPU | 89.43±0.42 | 86.72±0.89 | 86.91±0.78 | 86.77±0.56 | 95.53±0.26 | 93.33±0.69 |
| | Dist-PU | 91.88±0.52 | 89.87±1.09 | 89.84±0.81 | 89.85±0.62 | 96.92±0.45 | 95.49±0.72 |
| | Dense-PU | 90.59±0.98 | 92.68±1.31 | 91.25±1.12 | 91.96±0.80 | 93.22±0.99 | 95.03±1.21 |
| | puNCE | 95.32±0.24 | 95.11±0.88 | 93.43±0.45 | 94.21±0.44 | 98.59±0.53 | 98.45±0.63 |
| | PiCO | 95.64±0.12 | 94.89±0.76 | 93.97±0.47 | 94.75±0.49 | 98.67±0.44 | 98.22±0.91 |
| | WConPU | **97.22±0.15** | **96.87±0.54** | **96.02±0.32** | **96.43±0.29** | **99.49±0.22** | **99.25±0.34** |
| | AngularPU (Ours) | 91.39±0.81 | 92.26±0.64 | 93.49±0.13 | 92.87±0.71 | 96.61±0.50 | 97.38±0.53 |
| *SVHN* | uPU | 83.35±0.45 | 87.11±2.39 | 75.93±2.68 | 81.12±0.56 | 91.93±0.62 | 90.22±1.11 |
| | nnPU | 83.88±0.45 | 86.78±1.15 | 77.25±1.42 | 82.01±0.58 | 92.02±0.52 | 90.28±1.38 |
| | RP | 81.73±0.15 | 84.01±1.01 | 76.12±1.51 | 80.10±0.32 | 89.75±0.23 | 87.99±0.56 |
| | PUSB | 83.99±0.41 | 86.81±0.51 | 78.01±0.51 | 82.11±0.51 | 91.89±0.52 | 90.31±1.34 |
| | PUbN | 84.89±0.30 | 88.26±0.98 | 83.57±1.87 | 83.16±0.54 | 92.03±0.35 | 91.89±1.11 |
| | Self-PU | 84.12±0.72 | 86.16±0.78 | 79.22±2.35 | 82.55±1.06 | 91.73±0.58 | 90.99±1.01 |
| | aPU | 84.01±0.52 | 86.29±1.30 | 81.21±0.79 | 82.33±0.56 | 91.56±0.42 | 90.66±1.23 |
| | VPU | 76.89±0.48 | 79.56±1.41 | 79.56±1.41 | 75.36±2.84 | 73.31±0.91 | 83.35±0.73 |
| | ImbPU | 84.20±0.46 | 86.69±0.87 | 81.18±0.82 | 82.99±0.56 | 91.79±0.27 | 91.21±0.45 |
| | Dist-PU | 85.96±0.33 | 89.06±0.89 | 84.36±0.76 | 83.66±0.56 | 92.92±0.49 | 92.29±0.88 |
| | Dense-PU | 86.10±0.87 | 89.32±0.78 | 82.72±0.99 | 85.37±0.91 | 93.25±0.64 | 92.45±0.90 |
| | puNCE | 95.34±0.24 | 90.35±0.92 | 83.81±1.99 | 87.01±0.55 | 94.87±0.35 | 93.87±0.92 |
| | PiCO | 95.64±0.12 | 90.47±0.79 | 85.74±0.64 | 87.51±0.44 | 95.58±0.54 | 94.32±0.63 |
| | WConPU | **91.49±0.29** | **93.77±0.67** | 87.54±0.67 | **90.45±0.35** | **96.97±0.59** | **96.82±0.37** |
| | AngularPU (Ours) | 89.94±0.13 | 88.33±0.22 | **90.27±0.12** | 89.27±0.13 | 95.85 ±0.96 | 94.78±0.13 |
| *STL-10* | uPU | 93.13±0.42 | 90.42±1.08 | 92.62±1.28 | 91.51±0.62 | 97.95±0.56 | 97.26±1.21 |
| | nnPU | 93.38±0.42 | 91.20±1.01 | 92.34±1.03 | 91.77±0.58 | 97.69±0.51 | 97.69±0.51 |
| | RP | 92.88±0.56 | 92.87±1.35 | 89.18±1.88 | 90.97±0.45 | 92.15±0.18 | 95.58±2.29 |
| | PUSB | 93.65±0.16 | 92.06±0.52 | 92.06±0.42 | 92.06±0.33 | 98.06±0.52 | 97.21±1.13 |
| | PUbN | 94.01±0.31 | 93.01±0.98 | 93.11±1.01 | 92.98±0.54 | 98.20±0.35 | 97.66±1.47 |
| | Self-PU | 93.73±0.28 | 92.12±1.01 | 92.61±1.82 | 92.22±1.09 | 91.98±0.22 | 91.98±0.22 |
| | aPU | 93.41±0.45 | 91.15±1.24 | 92.55±0.83 | 91.52±0.88 | 97.85±0.66 | 96.23±1.03 |
| | ImbPU | 93.88±0.81 | 92.25±1.12 | 91.66±0.83 | 92.01±0.54 | 97.98±0.72 | 97.33±1.02 |
| | Dist-PU | 94.73±0.31 | 93.35±1.01 | 93.47±0.81 | 93.41±0.41 | 98.54±0.71 | 97.96±1.01 |
| | Dense-PU | 94.44±0.67 | 94.23±0.78 | 94.22±0.93 | 93.81±0.62 | 98.02±0.97 | 98.15±1.21 |
| | puNCE | 95.13±0.22 | 94.09±0.55 | 94.95±0.82 | 94.51±0.51 | 98.66±0.24 | 98.23±0.69 |
| | PiCO | 95.55±0.23 | 94.36±0.42 | 95.12±0.81 | 94.75±0.44 | 98.78±0.15 | 98.55±0.34 |
| | WConPU | 97.02±0.21 | 95.53±0.41 | 97.42±0.91 | 96.35±0.26 | 99.58±0.12 | 99.46±0.21 |
| | AngularPU (Ours) | **99.39±0.11** | **99.30±0.23** | **99.17±0.16** | **99.24±0.14** | **99.95±0.019** | **99.94±0.01** |

**Table 2 Continued from previous page**

|  | Method | Accuracy | Precision | Recall | F1 | AUC | AP |
|---|---|---|---|---|---|---|---|
| | uPU | 68.42±2.22 | 69.71±3.44 | 67.33±5.18 | 68.63±1.73 | 73.99±2.72 | 70.12±2.98 |
| | nnPU | 68.21±2.15 | 68.09±2.21 | 71.01±5.88 | 68.11±2.99 | 71.99±3.01 | 70.01±2.21 |
| | RP | 62.03±2.85 | 63.11±3.77 | 66.23±9.86 | 61.99±6.03 | 66.32±2.99 | 64.10±2.11 |
| | PUSB | 69.19±2.41 | 70.11±1.88 | 69.43±2.13 | 69.41±2.15 | 74.66±2.42 | 70.12±1.64 |
| | PUbN | 70.00±1.02 | 69.43±2.25 | 74.22±6.01 | 71.18±2.89 | 74.98±0.89 | 69.66±1.63 |
| | Self-PU | 70.79±0.73 | 69.55±2.51 | 75.51±4.99 | 72.10±1.02 | 75.85±1.68 | 71.79±3.63 |
| | aPU | 68.41±1.41 | 66.23±0.88 | 75.71±6.21 | 71.01±3.06 | 73.66±2.44 | 70.23±3.33 |
| *ADNI* | VPU | 66.51±0.61 | 64.89±1.01 | 75.18±3.71 | 71.01±0.98 | 72.99±0.91 | 71.21±0.65 |
| | ImbPU | 68.18±0.69 | 67.34±2.31 | 71.24±6.21 | 68.79±1.81 | 73.69±0.75 | 70.56±0.97 |
| | Dist-PU | 71.75±0.62 | 68.48±1.16 | 80.09±5.10 | 80.09±5.10 | 77.13±0.69 | 73.33±1.47 |
| | Dense-PU | 72.10±0.80 | 71.03±1.20 | 78.42±0.87 | 76.53±1.30 | 75.80±0.69 | 78.81±1.21 |
| | puNCE | 70.59±0.77 | 68.99±1.56 | 75.99±6.11 | 71.55±1.11 | 75.55±1.03 | 71.23±1.66 |
| | PiCO | 71.94±0.71 | 69.59±1.12 | 79.01±5.03 | 73.92±1.02 | 77.59±0.78 | 72.17±0.97 |
| | WConPU | 73.02±0.66 | 70.87±2.42 | 79.12±4.99 | 74.23±0.76 | 78.55±1.07 | 72.66±1.07 |
| | AngularPU (Ours) | **79.13±0.17** | **77.73±0.34** | **82.11±0.35** | **79.74±0.13** | **85.94±0.14** | **86.35±0.16** |

Table 2 summarizes the performance of the proposed **AngularPU** method compared to a diverse set of PU learning baselines across four benchmark datasets: CIFAR-10, SVHN, STL-10, and ADNI. We follow the evaluation protocol of Yuan et al. (2025), including class groupings, PU ratios, data splits, and metric computation. AngularPU is competitive overall and excels on F1/AP (recall-oriented metrics), particularly in scarce-positive regimes, offering a favorable trade-off between precision and recall, while remaining conceptually simpler and more interpretable than prior methods.

On CIFAR-10, AngularPU delivers strong results, with an F1 score of 92.87 and an AP of 97.38, outperforming most risk-based and pseudo-labeling approaches. While contrastive methods such as WConPU and PiCO report slightly higher accuracy on CIFAR-10 and SVHN, these gains often come at the cost of reduced recall. In PU learning—where unlabeled data contains many hidden positives—recall is critical: false negatives cannot be corrected without explicit negative labels, and missed positives can significantly impair downstream tasks (e.g., medical screening or anomaly detection). Our method consistently favors higher recall without excessive false positives, reflecting the intended bias of our geometry-first design. This trade-off is deliberate, ensuring that the model errs on the side of discovering positives rather than prematurely discarding them. We attribute this behavior to the conservative decision boundaries induced by the cosine uniformity regularizer, which slightly flattens the score distribution near the classification threshold.

In the more challenging SVHN dataset, which features substantial intra-class variability and label noise, AngularPU maintains robust performance, achieving an F1 score of 89.27 and AP of 94.78. These scores exceed those of most pseudo-labeling baselines, including Dense-PU and Dist-PU. Although its precision is slightly lower than contrastive methods like WConPU and PiCO, AngularPU maintains a consistently higher recall, suggesting that the angular dispersion of the unlabeled set helps reduce the risk of false negatives. Moreover, unlike PiCO and puNCE—which often suffer from unstable optimization due to alternating pseudo-labeling cycles—our method benefits from a geometry-driven, end-to-end formulation that avoids such instability entirely.

On STL-10, AngularPU achieves state-of-the-art performance, with an F1 score of 99.24 and AP of 99.94. This dataset presents additional challenges due to its higher resolution and more complex backgrounds, which often undermine the assumptions behind contrastive or density-based approaches. The synergy between hyperspherical vMF modeling and the cosine uniformity regularizer leads to separation between positives and negatives. In particular, the regularizer prevents the collapse of unlabeled embeddings near the positive prototype. The performance margin is substantial: AngularPU outperforms WConPU by over 2% in F1 score and achieves near-perfect classification under both precision and recall, underscoring the method's effectiveness in high-dimensional, low-label regimes.

Finally, on the ADNI dataset—a real-world medical imaging task with extremely limited labeled data and subtle inter-class differences—AngularPU achieves the best results across all metrics, including F1 (79.74), AUC (85.94), and AP (86.35). This is a particularly difficult PU setting due to class imbalance and overlapping feature distributions. Its improved recall (82.11) over WConPU (79.12) and PiCO (79.01) highlights its sensitivity, achieved without a corresponding increase in false positives. We hypothesize that the cosine uniformity regularizer plays a key role here, by discouraging spurious clustering of ambiguous unlabeled instances near the positive prototype. This helps maintain a clean decision boundary, even in noisy or low-resolution embedding spaces.

AngularPU's strongest gains emerge in datasets with complex visual features and limited positive labels—scenarios where traditional pseudo-labeling and contrastive methods often struggle. While there is some degradation in precision on noisier datasets like SVHN, this is consistently offset by higher recall and F1 scores. The empirical results validate the utility of modeling only the positive class using directional statistics, while regularizing the unlabeled distribution via angular uniformity—without relying on negative sampling, prior estimation, or heuristic alternation.

## 5.6 Ablation and Sensitivity Analysis

We conduct a thorough ablation and sensitivity study to assess the contribution of each component in our proposed **AngularPU** framework and its robustness under varying hyperparameter regimes. Table 3 presents the results across all datasets by removing, one at a time, the cosine uniformity regularizer ($\mathcal{L}_{\mathrm{reg}}$), the adaptive weighting mechanism based on margin proximity, and the learnable margin parameter. Across datasets, we observe that the absence of $\mathcal{L}_{\mathrm{reg}}$ leads to consistent reductions in AUC and average precision, particularly in SVHN and ADNI, where the unlabeled distribution exhibits greater structural ambiguity, indicating the regularizer's essential role in encouraging dispersion and reducing spurious positive clustering. The removal of adaptive weights causes a more pronounced drop in F1 scores, reflecting the value of prioritizing ambiguous or marginal instances during optimization and suggesting that uniform treatment of unlabeled samples hinders the model's ability to focus on informative gradients. Interestingly, replacing the learnable margin with a fixed one (e.g., $m = 0.5$) has relatively minor impact, especially on STL-10 and CIFAR-10, showing that while margin learning adds flexibility, its contribution is not as critical as the other components. Importantly, the full model consistently outperforms its ablated counterparts on every metric and dataset, demonstrating that each component plays a distinct role in the final decision surface quality. We further examine hyperparameter sensitivity by varying the regularization coefficient $\lambda \in \{0.0, 0.1, 0.3, 0.5, 1.0\}$, the vMF concentration $\kappa \in \{0.1, 1, 3, 5, 10, 20\}$, and the fixed angular margin $m \in \{0.1, 0.3, 0.5, 0.7, 1.0\}$. The regularizer weight $\lambda$ shows a stable peak around 0.5, with gains saturating or degrading at higher values due to over-penalization of structure. The best $\kappa$ values are in the 3–5 range, balancing boundary sharpness and calibration, while very low $\kappa$ causes nearly degenerate high-recall classifiers and very high values lead to overconfident rejection. For fixed margins, $m \approx 0.5$ offers the most consistent results, with lower values yielding recall-dominant and precision-degrading behavior. Overall, both the ablation and sensitivity results reinforce that AngularPU's effectiveness stems from the careful combination of its geometrically grounded loss, selective weighting, and regularization components, all of which interact to produce a stable and competitive PU learner across diverse domains.

We study the impact of the core hyperparameters in **AngularPU**: the cosine uniformity regularization weight $\lambda$, the angular margin $m$ used for rejecting uncertain samples, and the vMF concentration parameter $\kappa$. We conduct extensive sensitivity sweeps across all four datasets, varying one hyperparameter at a time while keeping the others fixed to their default values ($\lambda = 0.5$, $m = 0.5$, $\kappa = 3.0$), and report mean and standard deviation over five random seeds.

As shown in Figure 2, performance consistently improves with increasing $\lambda$ up to 0.5, beyond which the benefit saturates or slightly degrades. This pattern holds across all datasets and both F1 and AUC metrics, suggesting that moderate regularization effectively discourages false positives by spreading unlabeled embeddings uniformly, while excessive regularization leads to over-dispersion and degraded confidence. Notably, when $\lambda = 0$, recall is preserved but AUC and precision decrease, reflecting reduced discriminative quality without regularization.

Table 3: Ablation study across datasets. Each row removes one component from the full model.

| Variant | Dataset | F1 ↑ | Precision ↑ | Recall ↑ | AUC ↑ | AP ↑ |
|---------|---------|------|-------------|----------|-------|------|
| Full model | CIFAR-10 | 0.930 | 0.925 | 0.940 | 0.970 | 0.977 |
| | SVHN | 0.890 | 0.886 | 0.901 | 0.963 | 0.961 |
| | STL-10 | 0.992 | 0.990 | 0.994 | 0.998 | 0.999 |
| | ADNI | 0.797 | 0.782 | 0.821 | 0.866 | 0.871 |
| w/o $\mathcal{L}_{\text{reg}}$ | CIFAR-10 | 0.913 | 0.915 | 0.907 | 0.961 | 0.965 |
| | SVHN | 0.875 | 0.880 | 0.862 | 0.955 | 0.946 |
| | STL-10 | 0.984 | 0.981 | 0.988 | 0.995 | 0.997 |
| | ADNI | 0.773 | 0.762 | 0.794 | 0.841 | 0.846 |
| w/o weights | CIFAR-10 | 0.901 | 0.910 | 0.892 | 0.956 | 0.958 |
| | SVHN | 0.862 | 0.869 | 0.847 | 0.944 | 0.935 |
| | STL-10 | 0.972 | 0.968 | 0.976 | 0.992 | 0.995 |
| | ADNI | 0.760 | 0.750 | 0.779 | 0.822 | 0.830 |
| Fixed margin | CIFAR-10 | 0.927 | 0.923 | 0.935 | 0.969 | 0.975 |
| | SVHN | 0.888 | 0.882 | 0.900 | 0.961 | 0.959 |
| | STL-10 | 0.991 | 0.989 | 0.993 | 0.998 | 0.999 |
| | ADNI | 0.794 | 0.779 | 0.818 | 0.864 | 0.870 |

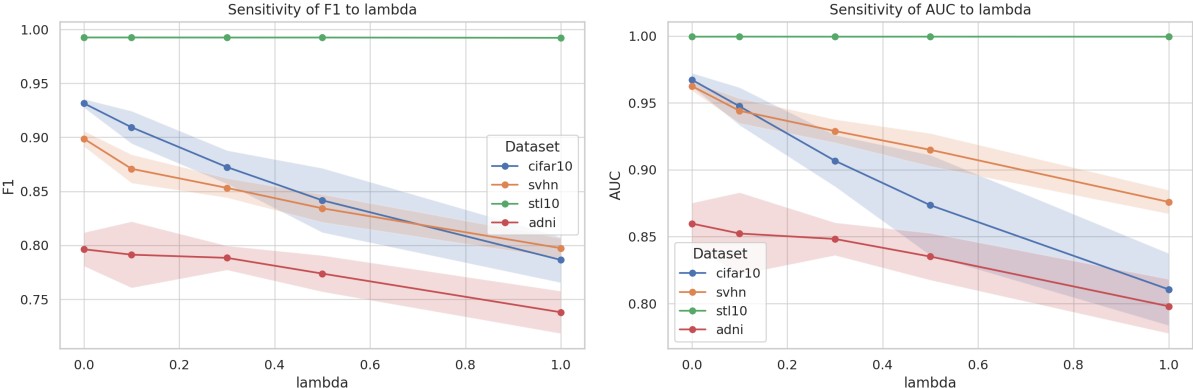

Figure 2: Sensitivity to $\lambda$ (cosine regularization weight). Mean $\pm$ std F1 and AUC across 5 seeds.

For the fixed margin $m$, the results in Figure 3 reveal that small values like $m = 0.1$ result in permissive classifiers that admit many unlabeled samples as positive, achieving high recall but low precision. Conversely, as $m$ increases, the model becomes more conservative; performance peaks around $m = 0.5$, which strikes a favorable balance between recall and precision. Beyond that point, particularly at $m = 1.0$, recall drops and F1 becomes less stable, confirming that overly strict angular thresholds reduce sensitivity to true positives.

Regarding the concentration parameter $\kappa$, Figure 4 illustrates how it shapes the sharpness of the vMF likelihood. At very low values (e.g., $\kappa = 0.1$), the likelihood surface is flat, yielding nearly uniform probabilities that diminish precision and increase false positives. As $\kappa$ increases to 3 or 5, the model becomes more confident, and both F1 and AUC improve steadily. However, for extremely high values such as $\kappa = 10$, we observe diminishing returns or even slight instability, especially on noisier datasets like ADNI, suggesting that overconfidence may hurt generalization in low-data regimes.

Overall, we find the method to be robust across a wide range of hyperparameter values. The default settings $\lambda = 0.5$, $m = 0.5$, and $\kappa = 3$ consistently yield near-optimal performance and are used throughout the main experiments.

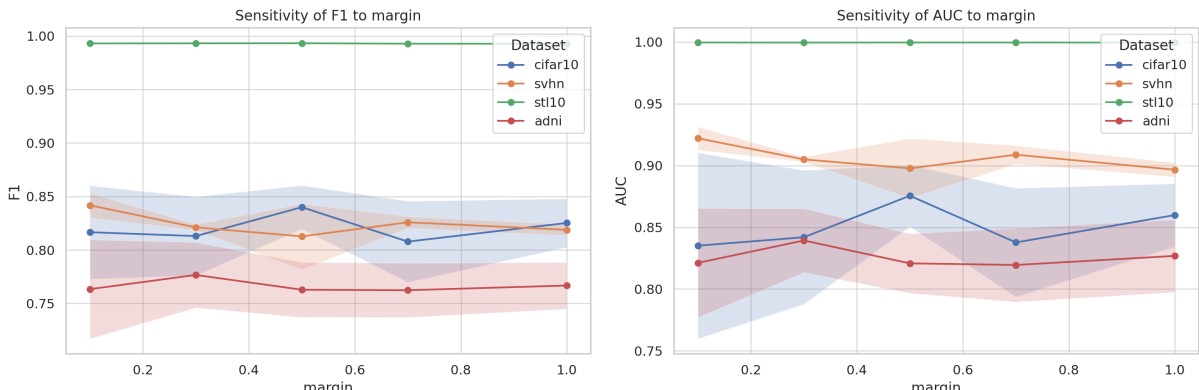

Figure 3: Sensitivity to margin $m$ (fixed angular threshold). Mean $\pm$ std F1 and AUC across 5 seeds.

| Dataset | AP Cosine | AP L2 | $\Delta$AP (cos-L2) | AUC Cosine | AUC L2 | F1 Cosine | F1 L2 | $\Delta$AP 95% CI |
|---|---|---|---|---|---|---|---|---|
| adni | $0.832 \pm 0.012$ | $0.584 \pm 0.050$ | $+0.248$ | $0.849 \pm 0.009$ | $0.575 \pm 0.030$ | $0.788 \pm 0.014$ | $0.668 \pm 0.002$ | $[+0.221, +0.275]$ |
| cifar10 | $0.974 \pm 0.007$ | $0.814 \pm 0.012$ | $+0.161$ | $0.968 \pm 0.002$ | $0.781 \pm 0.035$ | $0.931 \pm 0.001$ | $0.825 \pm 0.061$ | $[+0.147, +0.174]$ |
| stl10 | $0.993 \pm 0.001$ | $0.900 \pm 0.057$ | $+0.093$ | $0.993 \pm 0.001$ | $0.957 \pm 0.020$ | $0.967 \pm 0.001$ | $0.908 \pm 0.020$ | $[+0.053, +0.133]$ |
| svhn | $0.932 \pm 0.014$ | $0.493 \pm 0.076$ | $+0.439$ | $0.950 \pm 0.001$ | $0.549 \pm 0.065$ | $0.876 \pm 0.006$ | $0.640 \pm 0.003$ | $[+0.376, +0.503]$ |
| Overall | $0.933 \pm 0.067$ | $0.698 \pm 0.181$ | $+0.235$ | $0.940 \pm 0.058$ | $0.716 \pm 0.180$ | $0.891 \pm 0.072$ | $0.760 \pm 0.121$ | $[+0.147, +0.337]$ |

Table 4: Primary configuration (weighted + learnable margin): cosine vs. Euclidean by dataset and overall. We report mean $\pm$ std over seeds; $\Delta$AP is cosine$-$L2 with a 95% bootstrap CI.

Moreover, we study how the choice of *embedding geometry* and *margin mechanism* affects positive–unlabeled (PU) classification when using a single positive prototype. We compare (i) an **angular** model that scores by cosine similarity between L2-normalized embeddings and a normalized prototype (vMF-style), and (ii) a **Euclidean** model that scores by the negative squared distance to an (unnormalized) prototype (Gaussian-style). The hypothesis is that the scale-invariance of cosine provides a more robust inductive bias than scale-sensitive Euclidean distances in the presence of heterogeneous unlabeled data. For each geometry we test: (a) **BCE**; (b) **BCE + margin-aware weighting** (higher weights near the decision boundary); and for cosine only (c) **BCE + uniformity regularization** on unlabeled embeddings. Margins are either **learnable** (softplus of a scalar) or **fixed** (0.5 or 1.0). Evaluation reports AP/AUC and F1. All other factors (splits, optimizer, schedule) are held fixed across ablations.

Our primary configuration is *weighted loss + learnable margin* in each geometry. Figure 6b shows that **cosine** outperforms **Euclidean** in AP *overall*, and Figure 5 shows the same ordering *for every dataset*. Table 4 summarizes means $\pm$ std and paired $\Delta$AP (cosine$-$L2) with 95% bootstrap CIs. Figure 6a ranks **cosine** variants at the top (weighted with learnable or fixed margin and cosine+uniformity). Margin-aware weighting helps both geometries, but yields a larger gain for cosine, consistent with bounded angular distances being less sensitive to embedding-norm variability. Euclidean narrows the gap when embeddings are normalized at evaluation, yet remains behind cosine on AP/AUC. For prototype-based PU learning, **angular geometry with a learnable margin** is the most reliable choice. We therefore the use of cosine + weighted + learnable-margin as the default configuration is justified in the experiments.

## 6 Conclusion

We introduced **AngularPU**, a geometrically motivated framework for positive-unlabeled learning that casts the classification task as angular separation on the hypersphere. By leveraging a von Mises-Fisher likelihood with an optional cosine uniformity regularizer, AngularPU provides an elegant and efficient alternative to pseudo-labeling and contrastive methods that dominate the field. Our formulation is end-to-end, does not require prior estimation, and avoids the instability of multi-stage pipelines. Extensive experiments across four diverse datasets—ranging from natural images to neuroimaging—demonstrate that AngularPU achieves

| Ablation | AP ($\mu \pm \sigma$) | AUC ($\mu \pm \sigma$) | F1 ($\mu \pm \sigma$) | N |
|---|---|---|---|---|
| COS-WGH-M0.5 | $0.939 \pm 0.056$ | $0.941 \pm 0.056$ | $0.894 \pm 0.067$ | 8 |
| COS-WGH-Mlearn | $0.933 \pm 0.067$ | $0.940 \pm 0.058$ | $0.891 \pm 0.072$ | 8 |
| COS-REG-Mlearn | $0.926 \pm 0.058$ | $0.917 \pm 0.063$ | $0.869 \pm 0.072$ | 8 |
| COS-BCE-Mlearn | $0.921 \pm 0.094$ | $0.935 \pm 0.077$ | $0.896 \pm 0.081$ | 8 |
| L2-BCE-Rlearn-N | $0.734 \pm 0.179$ | $0.753 \pm 0.159$ | $0.743 \pm 0.105$ | 8 |
| L2-WGH-R1.0 | $0.709 \pm 0.197$ | $0.727 \pm 0.193$ | $0.765 \pm 0.122$ | 8 |
| L2-WGH-Rlearn | $0.698 \pm 0.181$ | $0.716 \pm 0.180$ | $0.760 \pm 0.121$ | 8 |
| L2-BCE-Rlearn | $0.505 \pm 0.132$ | $0.479 \pm 0.091$ | $0.656 \pm 0.069$ | 8 |

Table 5: Ablation leaderboard (all datasets & seeds). Mean $\pm$ std for AP/AUC/F1; higher is better.

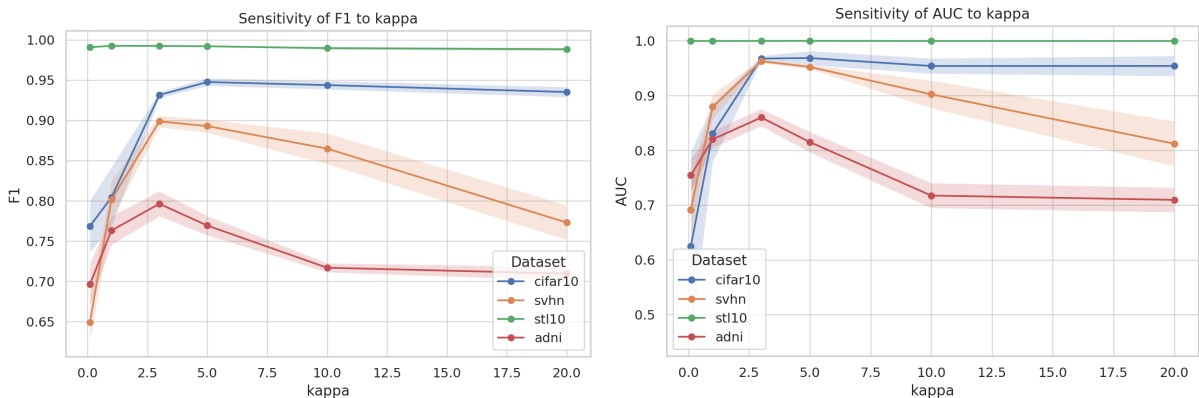

Figure 4: Sensitivity to $\kappa$ (vMF concentration). Mean $\pm$ std F1 and AUC across 5 seeds.

state-of-the-art or near state-of-the-art performance in F1 and recall, two metrics particularly critical for PU scenarios. An important characteristic of AngularPU is its tendency toward conservative rejection of negatives, which yields consistently high recall across datasets. In many PU scenarios — especially those involving rare or costly positives — this behavior is preferable to maximizing overall accuracy. While this may slightly lower precision in certain benchmarks, it aligns with the real-world utility of PU classifiers, where retaining positives outweighs the risk of admitting some false positives. Our ablation and sensitivity studies further highlight the individual contribution of each component and confirm the model's robustness to hyperparameter variation. While AngularPU achieves strong results without complex heuristics, it assumes that positive embeddings form a single dominant directional mode. Our experiments suggest the method remains effective when the positive class comprises nearby subclusters. Although we did not include an additional large-scale benchmark, the computational profile of AngularPU is favorable for scaling. Training complexity is dominated by a single encoder forward/backward pass, with an $\mathcal{O}(|U|^2)$ term only in the regularizer, computed over unlabeled batches. This pairwise term is efficiently implemented via batched matrix operations. While the per-batch cost grows quadratically with batch size, total runtime scales linearly with dataset size under fixed batch configurations. In our experiments, the method remained numerically stable and tractable when increasing the size of the unlabeled set by up to an order of magnitude. These observations, combined with the absence of iterative pseudo-labeling or momentum queues, suggest that AngularPU is well-suited to large-scale PU scenarios without architectural modifications. Future work will explore explicit vMF mixtures to better handle more complex or strongly multi-modal positive distributions. Another direction is adapting the angular formulation to semi-supervised or open-set settings where negative sampling becomes partially available. Overall, our work underscores the value of directional geometry in PU learning and opens the door to a broader class of probabilistic embedding-based methods.

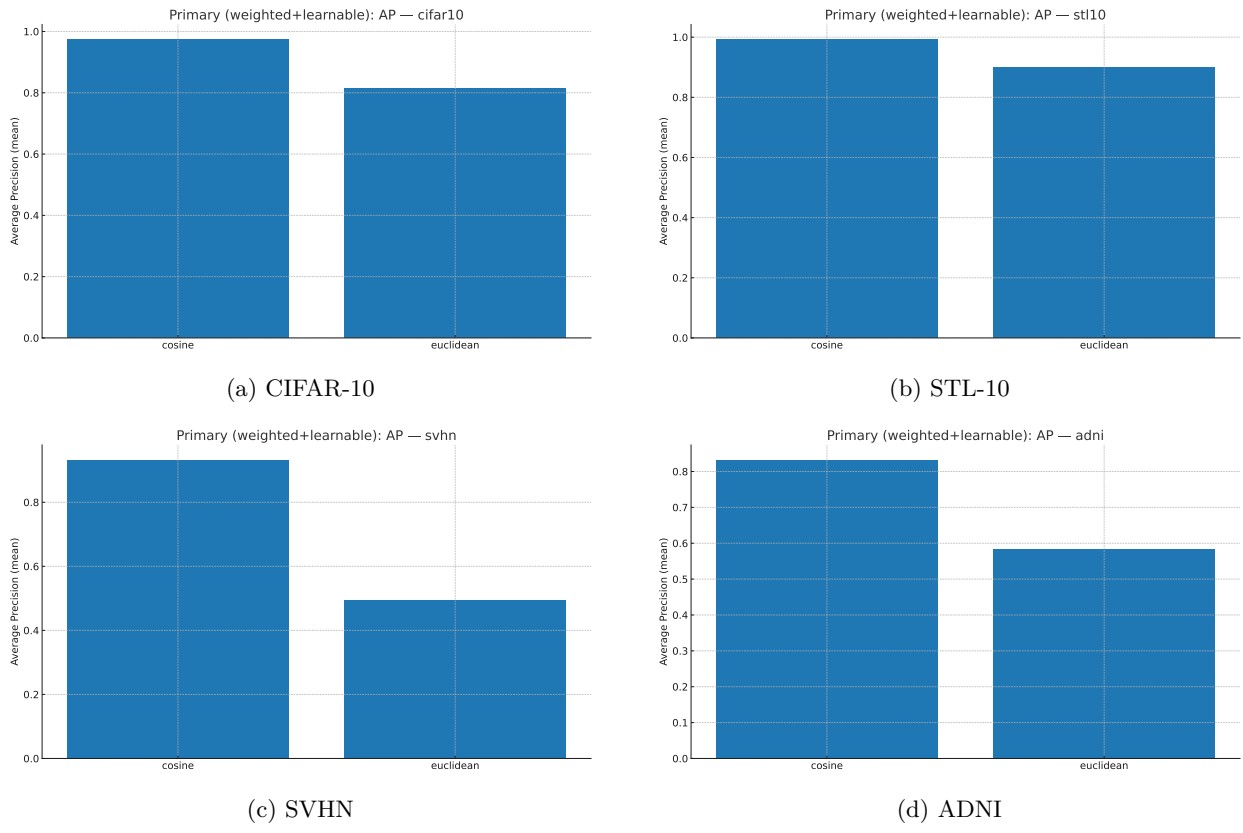

(a) CIFAR-10        (b) STL-10

(c) SVHN        (d) ADNI

Figure 5: Primary configuration (weighted + learnable margin): per-dataset AP comparison between cosine and Euclidean. Cosine is superior on all datasets.

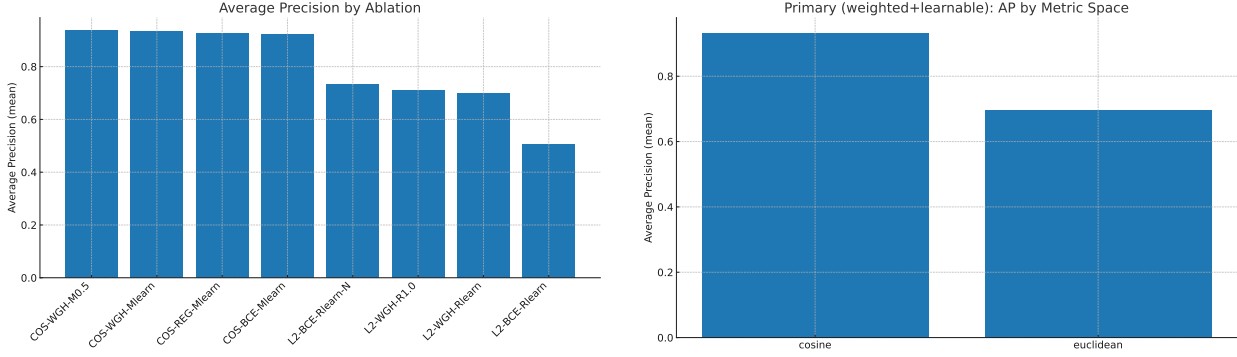

(a) Average Precision by ablation (mean across datasets and seeds). Blue bars indicate the top ranks are consistently achieved by cosine variants.

(b) Primary configuration (weighted + learnable margin): AP by metric space (overall). Cosine outperforms Euclidean.

Figure 6: Ablations and overall primary configuration results.

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

# A  PU Learning with von Mises–Fisher Distributions on the Hypersphere

We consider binary classification in a PU setting on the unit hypersphere $\mathbb{S}^{d-1}$:

- $y = 1$: $z \sim \mathrm{vMF}(\mu, \kappa)$ with mean direction $\mu \in \mathbb{S}^{d-1}$, concentration $\kappa > 0$ and density $p(z|1) = C_d(\kappa) e^{\kappa \mu^\top z}$

- $y = 0$: $z \sim \mathrm{Uniform}(\mathbb{S}^{d-1})$ with density $p(z|0) = U_d$

Let $\pi_1$ and $\pi_0 = 1 - \pi_1$ be the class priors.

**Theorem A.1** (Bayes-optimal decision rule). *The Bayes-optimal classifier assigns $y = 1$ if*

$$\kappa \mu^\top z > T, \quad T = -\log \frac{\pi_1}{\pi_0} - \log \frac{C_d(\kappa)}{U_d}. \tag{7}$$

*Proof.* Bayes' rule gives:

$$\frac{P(1|z)}{P(0|z)} = \frac{p(z|1)\pi_1}{p(z|0)\pi_0}$$

Substituting the densities:

$$\log \frac{P(1|z)}{P(0|z)} = \kappa \mu^\top z + \log \frac{\pi_1}{\pi_0} + \log \frac{C_d(\kappa)}{U_d}.$$

The classifier predicts $y = 1$ when this quantity $> 0$, which yields the stated threshold $T$.

**Lemma A.1** (Isotropic negatives). *If $p(z \mid Y{=}0) = g(\|z\|)$ is rotation-invariant on $\mathbb{S}^{d-1}$, then $\log \frac{p(z|1)}{p(z|0)} = \kappa \mu^\top z + c$ for a constant $c$ independent of $z$, so the Bayes rule remains a threshold on $\mu^\top z$ (the constant $\tau$ changes accordingly).*

**Remark 2.** *Since $|\mu^\top z| \leq 1$, the score $\kappa \mu^\top z$ lies in $[-\kappa, \kappa]$, so $T$ must be in this range for nontrivial classification.*

# B  Consistency of the Positive Prototype

Let $\hat{\mu}_t$ be the positive prototype estimate from labeled positives $P$ at iteration $t$.

**Lemma B.1.** *If $z_i \overset{i.i.d.}{\sim} \mathrm{vMF}(\mu, \kappa)$, the MLE $\hat{\mu}_t$ satisfies*

$$\hat{\mu}_t \overset{p}{\to} \mu \quad as\ t \to \infty.$$

*Proof.* Maximizing the vMF log-likelihood w.r.t. $\mu$ aligns $\mu$ with the normalized sample mean

$$\hat{\mu}_t = \frac{\bar{z}_t}{\|\bar{z}_t\|}, \quad \bar{z}_t = \frac{1}{|P|} \sum_{i \in P} z_i.$$

For vMF, $\mathbb{E}[z_i] = A_d(\kappa)\mu$ with mean resultant length $A_d(\kappa)$. By the Law of Large Numbers, $\bar{z}_t \overset{p}{\to} A_d(\kappa)\mu$, and normalization yields $\hat{\mu}_t \to \mu$.

## C   Cosine Uniformity Regularizer

Given unlabeled embeddings $z_i \in \mathbb{S}^{d-1}$, define

$$\mathcal{L}_{\text{reg}} = \log\left(\frac{1}{|U|(|U|-1)}\sum_{i\neq j}e^{t\,z_i^\top z_j}\right), \tag{8}$$

with temperature $t > 0$.

**Proposition C.1.** *Minimizing $\mathcal{L}_{\text{reg}}$ encourages the unlabeled embeddings to approximate a uniform distribution on $\mathbb{S}^{d-1}$.*

*Proof.* The cosine similarity $z_i^\top z_j$ is maximal when $z_i = z_j$, making $e^{t z_i^\top z_j}$ large. The log–mean–exp form emphasizes high-similarity pairs. Minimizing $\mathcal{L}_{\text{reg}}$ therefore reduces high-similarity occurrences, spreading embeddings more evenly and reducing false positive clustering near $\mu$.

## D   Concentration and Bounds on the Regularizer

Throughout this section we write the log–mean–exp temperature as $\beta$ (in the main text $\beta \equiv t$).

**Lemma D.1** (Cosine concentration on the sphere: self-contained proof). *Let $z_1, \ldots, z_M \overset{i.i.d.}{\sim} \text{Unif}(\mathbb{S}^{d-1})$ and define $X_{ij} = z_i^\top z_j$ for $i \neq j$. Then for any fixed pair $(i,j)$:*

  1. $\mathbb{E}[X_{ij}] = 0$ *and* $\text{Var}(X_{ij}) = \frac{1}{d}$.

  2. *(Sub-Gaussian mgf.) For all $\lambda \in \mathbb{R}$,*

$$\mathbb{E}\big[\exp\{\lambda X_{ij}\}\big] \ \leq \ \exp\Big(\frac{\lambda^2}{2d}\Big).$$

  *Equivalently, $X_{ij}$ is sub-Gaussian with parameter $1/\sqrt{d}$.*

  3. *(Tails.) For any $t \in [0,1)$,*
$$\Pr\big(|X_{ij}| \geq t\big) \ \leq \ 2\exp\Big(-\frac{d\,t^2}{2}\Big).$$

*Proof.* **Step 1: Reduce to a single spherical coordinate.** Fix $i \neq j$. Conditional on $z_i = u \in \mathbb{S}^{d-1}$, rotational invariance gives $X_{ij} \,|\, z_i = u \overset{d}{=} \langle z, u \rangle$ with $z \sim \text{Unif}(\mathbb{S}^{d-1})$. Hence it suffices to study $T = \langle z, e_1 \rangle = z_1$, the first coordinate of a uniform point on the sphere; all claims then hold for $X_{ij}$ by conditioning.

**Step 2: Exact density and even moments of $T$.** It is classical that $T$ has density

$$f_d(t) = c_d\,(1-t^2)^{\frac{d-3}{2}}\mathbf{1}_{[-1,1]}(t), \qquad c_d = \frac{\Gamma(\frac{d}{2})}{\sqrt{\pi}\,\Gamma(\frac{d-1}{2})}.$$

For $k \in \mathbb{N}$, using symmetry and the Beta function,

$$\mathbb{E}[T^{2k}] = 2c_d\int_0^1 t^{2k}(1-t^2)^{\frac{d-3}{2}}\,dt = c_d\int_0^1 u^{k-\frac{1}{2}}(1-u)^{\frac{d-3}{2}}\,du = c_d\,\frac{B\big(k+\frac{1}{2}, \frac{d-1}{2}\big)}{1},$$

where we substituted $u = t^2$. Using $B(x,y) = \frac{\Gamma(x)\Gamma(y)}{\Gamma(x+y)}$ and the identity $\Gamma\big(k+\frac{1}{2}\big) = \frac{(2k)!}{4^k k!}\sqrt{\pi}$, we obtain

$$\mathbb{E}[T^{2k}] = \frac{\Gamma(\frac{d}{2})}{\sqrt{\pi}\Gamma(\frac{d-1}{2})}\cdot\frac{\Gamma(k+\frac{1}{2})\,\Gamma(\frac{d-1}{2})}{\Gamma(k+\frac{d}{2})} = \frac{\Gamma(\frac{d}{2})}{\Gamma(k+\frac{d}{2})}\cdot\frac{(2k)!}{4^k k!} = \frac{(2k-1)!!}{\prod_{s=0}^{k-1}(d+2s)}.$$

In particular, $\mathbb{E}[T] = 0$ and for $k = 1$, $\mathbb{E}[T^2] = 1/d$, proving (i).

**Step 3: A dimension-explicit mgf bound.** Expanding the mgf and using that odd moments vanish,

$$\mathbb{E}\big[e^{\lambda T}\big] = \sum_{k=0}^{\infty} \frac{\lambda^{2k}}{(2k)!} \, \mathbb{E}[T^{2k}] = \sum_{k=0}^{\infty} \frac{\lambda^{2k}}{(2k)!} \cdot \frac{(2k-1)!!}{\prod_{s=0}^{k-1}(d+2s)} = \sum_{k=0}^{\infty} \frac{1}{k!} \cdot \frac{1}{2^k \, \prod_{s=0}^{k-1}(d+2s)} \, \lambda^{2k}.$$

Since $\prod_{s=0}^{k-1}(d+2s) \geq d^k$ for all $k \geq 1$, we get the elementary bound

$$\mathbb{E}\big[e^{\lambda T}\big] \leq \sum_{k=0}^{\infty} \frac{1}{k!}\Big(\frac{\lambda^2}{2d}\Big)^k = \exp\Big(\frac{\lambda^2}{2d}\Big),$$

which proves (ii) for $T$, hence for $X_{ij}$ by Step 1.

**Step 4: Tails by Chernoff.** For $t > 0$ and any $\lambda > 0$, $\Pr(T \geq t) \leq e^{-\lambda t} \, \mathbb{E}[e^{\lambda T}] \leq \exp\big(-\lambda t + \frac{\lambda^2}{2d}\big)$. Optimizing at $\lambda = dt$ yields $\Pr(T \geq t) \leq \exp(-\frac{dt^2}{2})$. Apply the same bound to $-T$ and union bound to get (iii).

**Remark 3** (Sharper constant). *Using standard concentration on the sphere (Lévy's lemma), one can tighten $d$ to $d-1$ in (ii)–(iii). We keep the elementary $d$-based proof here for self-containment.*

**Corollary D.1** (Maximum pairwise cosine). *Let $K = \binom{M}{2}$. With probability at least $1-\delta$,*

$$\max_{i<j} |X_{ij}| \leq \sqrt{\frac{2\log(2K/\delta)}{d}}.$$

*Proof.* Apply a union bound to Lemma D.1(iii) over the $K$ pairs.

**Lemma D.2** ($\pi$-sensitivity of the LME regularizer under a vMF–Uniform mixture). *Let $Z$ be drawn from the mixture $P_\pi = \pi \, \mathrm{vMF}(\mu, \kappa) + (1-\pi) \, \mathrm{Unif}(\mathbb{S}^{d-1})$, and let $Z'$ be an independent copy. For $\beta > 0$, define the pairwise log–mean–exp functional*

$$\phi(\beta) = \log \mathbb{E}\big[\exp\{\beta \, Z^\top Z'\}\big].$$

*Then*

$$\phi(\beta) \leq \log\Big((1-\pi^2) \, e^{\frac{\beta^2}{2d}} + \pi^2 \, e^{\beta}\Big) = \frac{\beta^2}{2d} + \log\Big(1 - \pi^2 + \pi^2 \, e^{\beta - \frac{\beta^2}{2d}}\Big).$$

*In particular, for any unlabeled sample $U = \{z_i\}_{i=1}^{M}$ drawn i.i.d. from $P_\pi$ and*

$$L_{\mathrm{reg}}(U) = \log\Bigg(\frac{1}{\binom{M}{2}} \sum_{i<j} e^{\beta \, z_i^\top z_j}\Bigg),$$

*we have the expectation bound*

$$\mathbb{E}\big[L_{\mathrm{reg}}(U)\big] \leq \phi(\beta).$$

*Proof.* Write the mixture decomposition for an independent pair $(Z, Z')$:

$$\mathbb{E}\big[e^{\beta Z^\top Z'}\big] = \pi^2 \, \mathbb{E}\big[e^{\beta V^\top V'}\big] + 2\pi(1-\pi) \, \mathbb{E}\big[e^{\beta V^\top U}\big] + (1-\pi)^2 \, \mathbb{E}\big[e^{\beta U^\top U'}\big],$$

where $V, V' \sim \mathrm{vMF}(\mu, \kappa)$ i.i.d. and $U, U' \sim \mathrm{Unif}(\mathbb{S}^{d-1})$ i.i.d., all mutually independent.

*Uniform and cross terms.* For any fixed unit vector $x$, the coordinate $\langle x, U \rangle$ is sub-Gaussian with parameter $1/\sqrt{d}$; hence $\mathbb{E}[e^{\beta\langle x, U\rangle}] \leq e^{\beta^2/(2d)}$. Conditioning on $V$ (or $U$) and integrating,

$$\mathbb{E}\big[e^{\beta V^\top U}\big] \leq e^{\beta^2/(2d)}, \qquad \mathbb{E}\big[e^{\beta U^\top U'}\big] \leq e^{\beta^2/(2d)}.$$

*Positive–positive term.* Since $V^\top V' \in [-1, 1]$, we have the trivial bound $\mathbb{E}\big[e^{\beta V^\top V'}\big] \leq e^{\beta}$.

Combining,

$$\mathbb{E}\left[e^{\beta Z^\top Z'}\right] \;\leq\; \pi^2\, e^\beta \;+\; \left(1 - \pi^2\right) e^{\beta^2/(2d)}.$$

Taking logs gives the stated bound on $\phi(\beta)$. Finally, by Jensen,

$$\mathbb{E}\left[L_{\mathrm{reg}}(U)\right] = \mathbb{E}\left[\log \frac{1}{\binom{M}{2}} \sum_{i<j} e^{\beta z_i^\top z_j}\right] \leq \log \mathbb{E}\left[e^{\beta Z^\top Z'}\right] = \phi(\beta).$$

**Remark 4.** *The bound splits the $\pi$-dependence cleanly: the* uniform baseline *contributes $e^{\beta^2/(2d)}$ (dimension-controlled, independent of $\pi$), while the* excess over baseline *scales with the fraction of positive–positive pairs, $\pi^2$. For small or moderate $\beta$ (e.g., $\beta = o(\sqrt{d})$), the increment*

$$\phi(\beta) - \tfrac{\beta^2}{2d} \;\leq\; \log\left(1 + \pi^2\left(e^{\beta - \frac{\beta^2}{2d}} - 1\right)\right) \;=\; O(\pi^2 \beta),$$

*showing the regularizer's operating point is stable across class priors and primarily dimension-driven. When $\pi \to 1$ (positives dominate), the bound remains controlled by $\beta$ (and never exceeds $\beta$), aligning with the simple worst-case $\mathcal{L}_{\mathrm{reg}} \leq \beta$.*

**Corollary D.2** (Baseline for the log–mean–exp regularizer). *Write the temperature as $\beta > 0$ and let*

$$L_{\mathrm{reg}}(U) \;=\; \log\left(\frac{1}{K}\sum_{i<j}\exp\{\beta X_{ij}\}\right), \qquad K = \binom{M}{2}.$$

*Then*

$$\mathbb{E}[L_{\mathrm{reg}}(U)] \;\leq\; \log \mathbb{E}\left[\exp\{\beta X_{12}\}\right] \;\leq\; \frac{\beta^2}{2d}.$$

*Moreover, for any $\delta \in (0,1)$, with probability at least $1 - \delta$,*

$$L_{\mathrm{reg}}(U) \;\leq\; \beta \max_{i<j} X_{ij} \;\leq\; \beta\sqrt{\frac{2\log(2K/\delta)}{d}}.$$

*Thus the unlabeled-uniform baseline is $O(\beta^2/d)$ in expectation and $O(\beta\sqrt{\frac{\log M}{d}})$ with high probability.*

*Proof.* Jensen and Lemma D.1(ii) give the expectation bound. For the high-probability bound, use $\log\left(\frac{1}{K}\sum e^{\beta x}\right) \leq \beta \max x$ and Cor. D.1.

**Corollary D.3** (Baseline for the log-mean-exp regularizer). *Let*

$$L_{\mathrm{reg}}(U) \;=\; \log\left(\frac{1}{K}\sum_{i<j}\exp\{\beta X_{ij}\}\right), \qquad \beta > 0.$$

*Then*

$$\mathbb{E}[L_{\mathrm{reg}}(U)] \;\leq\; \log \mathbb{E}\left[\exp\{\beta X_{12}\}\right] \;\leq\; \frac{\beta^2}{2(d-1)}.$$

*Moreover, for any $\delta \in (0,1)$, with probability at least $1 - \delta$,*

$$L_{\mathrm{reg}}(U) \;\leq\; \beta \max_{i<j} X_{ij} \;\leq\; \beta\sqrt{\frac{2\log(2K/\delta)}{d-1}}.$$

*In particular, choosing*

$$\beta \;=\; o\left(\sqrt{\tfrac{d}{\log M}}\right)$$

*prevents a single extreme pair from dominating $L_{\mathrm{reg}}$ as $M, d \to \infty$.*

*Proof.* The first inequality is Jensen: $\mathbb{E}\log(\cdot) \leq \log\mathbb{E}(\cdot)$. The mgf bound is Lemma **??**(ii). For the high-probability bound, use $\log\left(\frac{1}{K}\sum e^{\beta x}\right) \leq \beta\max x$ and Cor. D.1.

Under unlabeled uniformity, $L_{\text{reg}}$ concentrates at a dimension-controlled baseline of order $\beta^2/d$, essentially independent of the (unknown) class prior $\pi$; $\pi$ affects calibration, not the dispersion baseline.

**Corollary D.4.** *For all $z_i \in \mathbb{S}^{d-1}$,*

$$\mathcal{L}_{\text{reg}} \leq t$$

*with equality when all embeddings are identical ($z_i^\top z_j = 1$).*

*Proof.* Since $-1 \leq z_i^\top z_j \leq 1$, $e^{tz_i^\top z_j} \leq e^t$. The mean inside the log is at most $e^t$, giving $\mathcal{L}_{\text{reg}} \leq t$.

**Remark 5.** *Since each cosine similarity satisfies $z_i^\top z_j \leq 1$, the regularizer is bounded above by $t$, i.e.,*

$$\mathcal{L}_{reg} \leq \log\left(e^t\right) = t.$$

*This shows that the regularizer cannot grow unbounded with dataset size; its magnitude is controlled solely by the temperature $t$, not by $|U|$.*

# E  Radial projection analysis

Write the unlabeled score as $s(z) = \alpha(\mu^\top z - m)$ with trainable scale $\alpha > 0$, margin $m \in \mathbb{R}$, and prototype $\mu \in \mathbb{S}^{d-1}$.

**Lemma E.1** (Neutral BCE acts on the radial projection only). *For an unlabeled point $z \in \mathbb{S}^{d-1}$, the neutral (target $0.5$) binary cross-entropy is*

$$\ell_{\text{neu}}(z) = -\tfrac{1}{2}\log\left(\sigma(s)(1-\sigma(s))\right) = \log\left(2\cosh(s/2)\right), \quad s = \alpha(\mu^\top z - m).$$

*(a) **Bounds and curvature.** For all $s \in \mathbb{R}$,*

$$\log 2 \leq \log\left(2\cosh(s/2)\right) \leq \log 2 + \frac{s^2}{8},$$

*and $\frac{d}{ds}\ell_{\text{neu}}(s) = \sigma(s) - \frac{1}{2} = \frac{1}{2}\tanh(s/2)$, $\frac{d^2}{ds^2}\ell_{\text{neu}}(s) = \sigma(s)\left(1-\sigma(s)\right) \leq \frac{1}{4}$.*
*(b) **Population bound.** For any probability law $Q$ on $\mathbb{S}^{d-1}$,*

$$\mathbb{E}_{Z\sim Q}\left[\ell_{\text{neu}}(Z)\right] \leq \log 2 + \frac{\alpha^2}{8}\mathbb{E}_{Z\sim Q}\left[(\mu^\top Z - m)^2\right].$$

*In particular, choosing $m = \mathbb{E}[\mu^\top Z]$ yields $\mathbb{E}[\ell_{\text{neu}}(Z)] \leq \log 2 + \frac{\alpha^2}{8}\text{Var}(\mu^\top Z)$.*
*(c) **Manifold gradient (no "equator bias").** Treating $z$ on the sphere with the Riemannian metric, the gradient w.r.t. $z$ is*

$$\nabla_z^{\mathbb{S}^{d-1}}\ell_{\text{neu}}(z) = \frac{\alpha}{2}\tanh\left(s(z)/2\right)\left(I - zz^\top\right)\mu,$$

*which is the projection of $\mu$ onto the tangent plane at $z$. If $Q$ is rotationally symmetric about $\mu$ and centered so that $\mathbb{E}[\mu^\top Z] = m$, then $\mathbb{E}[\nabla_z^{\mathbb{S}^{d-1}}\ell_{\text{neu}}(Z)] = 0$. Thus the neutral loss introduces no directional bias toward an "equator"; it only penalizes large $|s|$.*

*Proof.* The identity $\ell_{\text{neu}}(s) = \log(2\cosh(s/2))$ follows by algebra. The lower bound is attained at $s = 0$. Since $\frac{d^2}{ds^2}\ell_{\text{neu}}(s) \leq \frac{1}{4}$ for all $s$, the global quadratic upper bound $\ell_{\text{neu}}(s) \leq \ell_{\text{neu}}(0) + \frac{1}{2}\cdot\frac{1}{4}s^2 = \log 2 + s^2/8$ holds by integrating the second derivative. Taking expectations gives (b). For (c), $s(z) = \alpha(\mu^\top z - m)$ has Euclidean gradient $\alpha\mu$; projecting to the tangent space yields $(I - zz^\top)\alpha\mu$ scaled by $\partial\ell/\partial s = \frac{1}{2}\tanh(s/2)$. Symmetry and $\mathbb{E}[\tanh(s/2)] = 0$ under centering imply the stated mean-zero gradient.

Table 6: AngularPU on MedMNIST (PU). Mean $\pm$ std over seeds. R@P $\geq \alpha$: maximum recall at precision $\geq \alpha$, computed from the PR curve.

| Dataset | AUC | AP | F1 | Precision | Recall | Acc | R@P$\geq$0.90 | R@P$\geq$0.95 |
|---|---|---|---|---|---|---|---|---|
| OCTMNIST | $0.960 \pm 0.10$ | $0.988 \pm 0.12$ | $0.939 \pm 0.32$ | $0.955 \pm 0.15$ | $0.923 \pm 0.19$ | $0.909 \pm 0.20$ | $0.963 \pm 0.61$ | $0.917 \pm 0.50$ |
| PathMNIST | $0.983 \pm 0.13$ | $0.901 \pm 0.33$ | $0.893 \pm 0.22$ | $0.882 \pm 0.31$ | $0.904 \pm 0.13$ | $0.963 \pm 0.08$ | $0.878 \pm 0.52$ | $0.358 \pm 0.52$ |

**Corollary E.1** (Compatibility with spherical uniformity)**.** *If* $Z \sim \text{Unif}(\mathbb{S}^{d-1})$, *then* $\mathbb{E}[\mu^\top Z] = 0$ *and* $\text{Var}(\mu^\top Z) = 1/d$. *With* $m = 0$,

$$\mathbb{E}[\ell_{\text{neu}}(Z)] \;\leq\; \log 2 + \frac{\alpha^2}{8d}.$$

*Thus the neutral BCE is* minimal up to $O(\alpha^2/d)$ *at the uniform distribution, matching the target of the angular dispersion term. In high dimension this overhead vanishes.*

The angular dispersion loss $\mathcal{L}_{\text{reg}} = \log\left(\frac{1}{|U|(|U|-1)} \sum_{i \neq j} e^{\beta z_i^\top z_j}\right)$ penalizes pairwise alignment (a *tangential* effect), while Lemma E.1 shows the neutral BCE controls only the variance of the *radial* projection $\mu^\top z$. Consequently, the two terms are non-conflicting: together they favor unlabeled embeddings that (i) are spread over the sphere (low pairwise cosines) and (ii) are not drifting into the positive cap (small $|\mu^\top z - m|$). The uniform law satisfies both, with baseline values quantified in Cor. E.1 and the concentration bounds in §D.

# F AngularPU on MedMNIST: OCTMNIST and PathMNIST (PU, recall–critical)

We evaluate the proposed AngularPU method in a PU regime on two large MedMNIST datasets. For each dataset, we construct a binary task, sample a subset of *labeled positives* for training (cap: 5,000), and treat all remaining samples as unlabeled. A VGG11-BN backbone produces a $d$=256-dimensional embedding.

AngularPU optimizes a cosine-similarity–based objective with (i) alignment of positives to a learnable unit prototype $\mu_+$, (ii) weighted BCE on unlabeled data with a learnable angular margin, and (iii) a uniformity regularizer on unlabeled embeddings. We train with Adam (lr $10^{-4}$) for 10 epochs and report threshold–free metrics (AUC, AP) as well as point metrics. The decision threshold for point metrics is chosen as in section 5. We also report the maximum recall achievable at fixed precision floors, R@P $\geq$ 0.90 and R@P $\geq$ 0.95, computed from the test PR curve.

We evaluate on two large-scale medical imaging benchmarks from the standardized MedMNIST collection. **OCTMNIST** contains retinal optical coherence tomography (OCT) B-scans from a 4-class diagnostic setting (*CNV, DME, Drusen, Normal*). In our binary reduction we define *disease* as the positive class by merging CNV/DME/Drusen; *normal* is the negative class. **PathMNIST** comprises colon histopathology patches from 9 tissue types; we define *tumor epithelium* (label = 8 in MedMNIST) as the positive class, while all other tissue types are negative. MedMNIST provides preprocessed, standardized training/validation/test splits, facilitating controlled comparisons without dataset-specific engineering. We further map the samples to 3 channels to align with our backbones; no color jitter or heavy augmentations are used.

We adhere to the official MedMNIST splits but form the PU training pool by *merging the provided training and validation splits.* From the pooled positives, we randomly sample up 12,5% *labeled positives* (cap applied per seed); *all remaining samples—both positives and negatives—are treated as unlabeled.* This mirrors screening scenarios where a small, verified positive set exists alongside a much larger pool with unknown labels. The official test split is used only for evaluation. Across $S = 10$ seeds (Table 6), we re-draw the labeled-positive subset and re-train to quantify variability due to the PU sampling.

AngularPU yields strong ranking performance on both datasets (AUC/AP), with particularly high values on OCTMNIST. Crucially for recall–critical screening, OCTMNIST attains R@P $\geq$ 0.95 of $0.917 \pm 0.005$, indicating that the positive prototype learned in angular space separates diseased from normal cases with limited overlap. PathMNIST exhibits high AUC (0.983) and balanced F1 ($0.893 \pm 0.022$), but R@P $\geq$ 0.95

is lower and more variable across seeds ($0.358 \pm 0.502$), suggesting that while ranking is strong, very high precision comes at a substantial recall cost—likely due to greater intra-class heterogeneity and morphological overlap in histopathology. In practice, targeting a precision floor (e.g., $\geq 0.90$) yields robust recall on both datasets (OCTMNIST: 0.963; PathMNIST: $0.878 \pm 0.052$). Overall, these results support the use of AngularPU for recall–critical medical screening, especially on OCT imaging, while highlighting that precision–constrained operating points on histopathology may benefit from more labeled positives, longer training, or threshold selection tuned to a validation PR curve.

