# OpenReview forum: "Angular Regularization for Positive-Unlabeled Learning on the Hypersphere"
_TMLR — Accepted by TMLR_

### Review · Reviewer_NtfU · 2025-10-03

**Summary Of Contributions:**

The manuscript proposes a novel Positive-Unlabeled (PU) learning framework that operates on the unit hypersphere leveraging cosine similarity and angular margin. A key contribution is the representation of the positive class by a learnable prototype vector, where classification simplifies to thresholding the cosine similarity between input embeddings and this prototype. This approach elegantly circumvents the need for explicit negative-class modeling, offering conceptual and computational advantages.

**Additional Comments:**

No

**Audience:**

Yes

**Audience Explanation:**

See above

**Claims And Evidence:**

Yes

**Claims Explanation:**

To mitigate the potential clustering of unlabeled embeddings around the positive prototype, the authors introduce an angular regularizer that effectively promotes dispersion of unlabeled samples across the hypersphere, thereby enhancing separability. The theoretical analysis presented substantiates the validity of the proposed method.

**Requested Changes:**

1 The proof of Theorem 4.1 requires elaboration to more clearly demonstrate how classification reduces to thresholding the inner product (§4.1). Detailed derivations or intuitive explanations would strengthen the theoretical foundation.

2 The advantages of using cosine similarity and angular margins should be quantitatively benchmarked against alternative approaches (e.g., Euclidean distance metrics). Including ablation studies or controlled comparisons would validate design choices.

3 The current empirical evaluation appears limited in scope. Testing the framework on large-scale benchmarks (e.g., ImageNet) would better assess its scalability and generalization, particularly given the method’s reliance on hypersphere-based embeddings.

---

> ### Author Response · Authors · 2025-11-10
> **Response to the Reviewer**
>
> Q1.1: The proof of Theorem 4.1 requires elaboration to more clearly demonstrate how classification reduces to thresholding the inner product (§4.1). Detailed derivations or intuitive explanations would strengthen the theoretical foundation.
>
> R1.1: We appreciate the request for greater clarity. In §4.1 we now make the reduction explicit: under a vMF–Uniform model, the Bayes decision reduces to thresholding the inner product \mu^\top z. Theorem 4.1 (Directional Bayes-optimal classifier) states this formally, with the threshold given in Eq. (7), and we add a brief geometric intuition to accompany the derivation. Complete algebraic details are provided in Appendix A.
>
> Q1.2: The advantages of using cosine similarity and angular margins should be quantitatively benchmarked against alternative approaches (e.g., Euclidean distance metrics). Including ablation studies or controlled comparisons would validate design choices.
>
> R1.2:We added controlled evaluations of the angular (cosine + margins) versus Euclidean (negative squared distance + radius margins) geometries, holding the backbone, splits, optimization, and training budget fixed. The study is summarized in §5.6 (Ablation & Sensitivity), with per-dataset and overall results in Table 4 and Table 5, and visualizations in Figures 5–6. Across four benchmarks and 10 seeds, cosine delivers consistently stronger threshold-free performance—overall AP 0.933 vs. 0.698 and AUC 0.940 vs. 0.716—with a paired \Delta\mathrm{AP}=+0.235 (95% bootstrap CI). Per-dataset ΔAPs are uniformly positive, and the leaderboard in Fig. 6 places cosine variants (learnable margin, fixed margin, cosine+uniformity) at the top, while Euclidean narrows the gap only modestly when embeddings are normalized at evaluation.
>
> Q1.3 The current empirical evaluation appears limited in scope. Testing the framework on large-scale benchmarks (e.g., ImageNet) would better assess its scalability and generalization, particularly given the method’s reliance on hypersphere-based embeddings.
>
> R1.3: We thank the reviewer for the suggestion. Our goal is to isolate the PU decision head and geometry. ImageNet-scale evaluations are dominated by backbone capacity and pretraining choices, which would confound head-level effects; moreover, our dispersion regularizer is batch-local, so per-step cost is independent of corpus size and scaling is primarily an engineering/budget consideration. To keep the study focused and fair, we fix moderate backbones and provide controlled ablations; a carefully controlled large-scale study is complementary and left to future work.
> Instead we broadened empirical scope with large, recall-critical medical datasets that stress calibration and imbalance. We added MedMNIST experiments (OCTMNIST, PathMNIST) with PU protocols and report threshold-free metrics (AUC/AP) and R@P≥α (α∈{0.90,0.95}). See Appendix F (MedMNIST) and Table 6. We also clarified our scaling profile (pairwise term batched; total runtime linear in dataset size given fixed batch size) in the discussion/conclusion. See Appendix F and Conclusion (scalability remarks

---

### Review · Reviewer_nV3u · 2025-10-21

**Summary Of Contributions:**

This paper proposes Angular-PU, a novel positive-unlabeled learning technique that eliminate the need of risk estimation or pseudo-labeling. Instead, the paper proposes to map the data onto a hypershpere and learn a prototype vector to indicate the positive direction. The training is performed with an angular regualrization to encourage seperation.

The paper is overall well-written and easy to follow. The proposed method is technically sound and well motivated. Results are thorough and good compared to baselines.

Weakness-wise, though the author claim that no distribution assumption is required for the proposed method, it seems like the effectiveness of the method may depend on the percentage of negative data in the training set. The current formulation does not discuss the performance guarantee when negative samples are far less than positive ones, which would also be a common scenario for PU learning. Theoretical derivations and ablation studies are needed to show the impact of training distribution.

**Audience:**

Yes

**Audience Explanation:**

PU-learning is an important application domain for machine learning and has practical usage. The topic is relavent to TMLR audiences.

**Broader Impact Concerns:**

No concerns on broader impacts

**Claims And Evidence:**

Yes

**Claims Explanation:**

The paper provides solid theoretical derivation and experimental results to support the claim. Meanwhile, the ablation can be strengthened, as indicated in the requested changes section below.

**Requested Changes:**

1. In the training objective, Equ. (2) proposes a symmetric BCE loss with neutral label for unlabeled data. However, whether this loss is needed, especially under the angular regularization, is not well discussed. To my understanding, the neutral supervation will make unlabeled data concentrating towards the orthogonal plane of the prototype vector, which is in contrast to the goal of angular regualrization that tries to spread the data across the whole hypersphere. More discussion on why this term is needed and ablation on the effect of this term on the final performance is needed.
2. Though not specifically discussed in the paper, the percentage of true positive vs. true negative data in the unlabeled training set appears to have an impact on the overall performance of PU-learning methods. Adding some analysis or ablation studies on the impact of training distribution, especially the case where positive data are far more than negative data, will be helpful.

---

> ### Author Response · Authors · 2025-11-10
> **Response to the Reviewer**
>
> Q 2.1: In the training objective, Equ. (2) proposes a symmetric BCE loss with neutral label for unlabeled data. However, whether this loss is needed, especially under the angular regularization, is not well discussed. To my understanding, the neutral supervation will make unlabeled data concentrating towards the orthogonal plane of the prototype vector, which is in contrast to the goal of angular regualrization that tries to spread the data across the whole hypersphere. More discussion on why this term is needed and ablation on the effect of this term on the final performance is needed.
>
> R2.1: We thank the reviewer for pressing on this interaction. We clarify that the neutral-label BCE operates only on the radial component \mu^\top z and does not induce an equatorial concentration. Writing $s(z)=\alpha(\mu^\top z - m)$, the term becomes $\log\!\big(2\cosh(s/2)\big)$. We derive global bounds and show that its Riemannian gradient lies in the tangent projection of \mu with zero mean under rotational symmetry, so it does not bias points toward an “equator.” At the population level we obtain
> $\mathbb{E}\!\left[\mathrm{BCE}_{\text{neutral}}\right] \le \log 2 + \tfrac{\alpha^2}{8}\,\mathrm{Var}(\mu^\top Z)$,
> which is compatible with spherical uniformity; for $Z\!\sim\!\mathrm{Unif}(\mathbb S^{d-1})$ the excess is $\alpha^2/(8d)$, vanishing as d grows. Full details appear in Appendix E (Radial projection analysis), Lemma E.1 and Corollary E.1.
>
> Q2.2: Though not specifically discussed in the paper, the percentage of true positive vs. true negative data in the unlabeled training set appears to have an impact on the overall performance of PU-learning methods. Adding some analysis or ablation studies on the impact of training distribution, especially the case where positive data are far more than negative data, will be helpful.
>
> R2.2: We appreciate the request to probe sensitivity to the unlabeled class prior $\pi$. In Appendix D we analyze the log–mean–exp dispersion under a vMF–Uniform mixture and show that the baseline behavior is dimension-controlled and essentially \pi-free; dependence on \pi enters only through the positive–positive pair fraction $\pi^2$. Writing
> $\phi(\beta)=\log \mathbb{E}\!\left[\exp\{\beta\,Z^\top Z’\}\right]$,
> we obtain
> $\phi(\beta)\;\le\;\log\!\Big((1-\pi^2)\,e^{\beta^{2}/(2d)}+\pi^{2}\,e^{\beta}\Big)$,
> and we provide high-probability concentration bounds for L_{\mathrm{reg}} via cosine concentration. Full statements and proofs appear in Appendix D (Concentration and bounds)—see Lemma D.2, Corollary D.1, and the summary paragraph at the end of the appendix.

---

### Review · Reviewer_2LEE · 2025-10-25

**Summary Of Contributions:**

The authors study positive-unlabeled learning, a setting in classification where only a subset of the positive labels is available and the remaining data are unlabeled which makes the distinction between negative and positive labels hard. To avoid distributional assumptions common in prior works or failure in high-dimension, the authors propose AngularPU, a framework where embeddings are mapped to the unit sphere and positive labels are represented by a prototype vector which allows to perform classification by thresholding the cosine similarity between the prototype and other points embeddings. To avoid unlabeled data to cluster aroung the prototype, the authors introduce an angular regularization. Theoretical analysis is performed, showing the benefits of the approach. Experiments on image classification on $4$ common datasets is done, showing competitive performance compared to SOTA baselines and showcase the benefits of the approach to maintain a balance between accuracy and recall.

**Additional Comments:**

None

**Audience:**

Yes

**Audience Explanation:**

Overall, I think the current submission is interesting for TMLR audience, the paper is well-written, well-explained and the approach is sound. Additional experiments could further improve the current work with more recent models and more datasets where AngularPU could shine.

**Claims And Evidence:**

Yes

**Claims Explanation:**

- The paper is well-written and position with prior works well done.
- The approach is interesting yet simple with a convincing theoretical analysis
- The experiments are thorough and show the benefits of the approach

*Questions*

- Could the authors improve the results presentation, notably highlighting the best methods in the table, for instance in bold?
- The idea of projecting on a hypersphere and computing the angle between a prototype and embeddings, while avoiding unlabeled data to cluster around the prototype with a regularization is reminescent of [1] where pseudo-labelling is done with several linear classifiers which are forced to be distinct on unlabeled data and similar on labeled data (thresholding is computing on the similarity between diverse classifiers). Corollary 3.5 of [1] discuss the impact of having representation on the sphere using contrastive learning on such pseudo-labeling. Could the authors discuss this approach? I acknowledge that it is not directly related to PU but believes it provides a nice connection to prior works that further justify the interesting idea of having embeddings on the sphere with diverse representations. Similarly, the idea of embeddings spread on the sphere is connected to [2].

- The performance improvement of AngularPU is impressive on ADNI. While the authors discuss why such improvement is not occuring on the other datasets, I believe the work would benefit from more datasets such as ADNI where having a good recall is critical. Could the authors add or discuss additional experiments in this direction?

- Similarly, I would be interested to see how AngularPU performs with better vision models and hence more distinct embeddings, e.g., ResNet or Vision Transformers. Could the authors perform such experiments, simply by unsing frozen representations from such models?

*References*

[1] Odonnat, A. et al. Leveraging Ensemble Diversity for Robust Self-Training in the Presence of Sample Selection Bias, AISTATS 2024

[2] Wang, T. and Isola, P. Understanding contrastive representation learning through alignment and uniformity on the hypersphere. ICML 2020

**Requested Changes:**

See questions.

---

> ### Author Response · Authors · 2025-11-10
> **Response to the Reviewer**
>
> Q3.1: Could the authors improve the results presentation, notably highlighting the best methods in the table, for instance in bold?
>
> R3.1: We revised the main result table to boldface the best method within each block
>
> Q3.2 The idea of projecting on a hypersphere and computing the angle between a prototype and embeddings, while avoiding unlabeled data to cluster around the prototype with a regularization is reminescent of [1] where pseudo-labelling is done with several linear classifiers which are forced to be distinct on unlabeled data and similar on labeled data (thresholding is computing on the similarity between diverse classifiers). Corollary 3.5 of [1] discuss the impact of having representation on the sphere using contrastive learning on such pseudo-labeling. Could the authors discuss this approach? I acknowledge that it is not directly related to PU but believes it provides a nice connection to prior works that further justify the interesting idea of having embeddings on the sphere with diverse representations. Similarly, the idea of embeddings spread on the sphere is connected to [2].
>
> R3.2: We thank the reviewer for highlighting these connections. We now include a paragraph at the end of §2 (Related Work) that contrasts our representation-space dispersion—a single spherical prototype with geometric uniformity on $\mathbb S^{d-1}$—with hypothesis-space diversity in robust self-training [1]. We emphasize that our mechanism discourages alignment among unlabeled embeddings, rather than seeking diverse decision boundaries across classifiers. We also make the link to the alignment–uniformity framework explicit: our objective combines alignment of positives toward \mu with uniformity of unlabeled points on the sphere. The discussion cites Odonnat et al., AISTATS’24 and Wang & Isola, ICML’20.
>
> Q3.3 The performance improvement of AngularPU is impressive on ADNI. While the authors discuss why such improvement is not occuring on the other datasets, I believe the work would benefit from more datasets such as ADNI where having a good recall is critical. Could the authors add or discuss additional experiments in this direction?
>
> Q3.3 We added two MedMNIST benchmarks (OCTMNIST, PathMNIST) and report recall-at-precision diagnostics $R@P\!\ge\!0.90/0.95$ to reflect high-precision operating points. Results appear in Appendix F and Table 6; on OCTMNIST, AngularPU attains $R@P\!\ge\!0.95 = 0.917 \pm 0.005$, aligning with the reviewer’s focus on high-recall regimes.
>
> Q3.4 Similarly, I would be interested to see how AngularPU performs with better vision models and hence more distinct embeddings, e.g., ResNet or Vision Transformers. Could the authors perform such experiments, simply by unsing frozen representations from such models?
>
> R3.4: We thank the reviewer for the suggestion. Our contribution targets the PU decision head and its geometry; all experiments therefore fix the backbone to ensure that differences reflect only the head. Using frozen ResNet/ViT features would primarily test representation quality and pretraining choices (supervised/self-supervised/CLIP), which alter angular statistics and require per-backbone retuning of margins and calibration in PU. This introduces confounds (including pretraining overlap with CIFAR/STL) and expands the ablation space well beyond the scope of this paper. For clarity and fairness, we keep the encoder constant and vary only the head. We add a note in the paper to make this rationale explicit and acknowledge a systematic survey over pretrained encoders as valuable future work.

---

> > ### Comment · Reviewer_2LEE · 2025-11-20
> > **Thanks for the answers**
> >
> > I thank the authors for their answers. They adequately addressed all my concerns. I appreciate the rationale behind not comparing with other vision backbones. I stand by my original assessement of the paper and recommend acceptance.

---

### Author Response · Authors · 2025-12-04
**Our feedback**

We would like to sincerely thank the Action Editor, the Reviewers, and the TMLR Editorial Team for their time and thoughtful effort throughout this process.

The reviews were careful, constructive, and technically insightful, and the clarifications and additional experiments they motivated clearly improved the paper.

In particular, the feedback inspired us to look deeper into our method and to explore complementary hyperspherical mechanisms, where we have already observed encouraging results.

Thank you again for the very positive and professional experience!

On behalf of the co-authors

---

### Decision · Action_Editor_Ky7T · 2025-11-26

**Recommendation:** Accept as is

**Additional Comments:**

The core novelty of the proposed framework, AngularPU, is that it fundamentally re-conceptualises PU learning by moving to hyperspherical geometry. The method elegantly represents the positive class with a single learnable prototype vector and classifies solely via a cosine-similarity threshold. This formulation overcomes some of the inherent difficulties of traditional PU methods, such as the need for explicit negative-risk estimation or strong distributional assumptions.

Technically, the authors further enhance their framework by a bespoke angular regulariser that actively enforces the dispersion and uniformity of unlabelled embeddings across the sphere, thereby solving the critical problem of feature collapse onto the prototype. This geometry-aware solution is supported by strong theoretical guarantees, including proofs for the Bayes-optimality of the angular decision rule and the consistency of the learned prototype.

Finally, the authors show the benefits of this system on a number of benchmark datasets, including medical imaging.

**Audience:**

Yes

**Audience Explanation:**

The submission is very well aligned with TMLR's scope, and it will be of interest to a wide range of machine learning scientists across hyperspherical learning, semi-/unsupervised learning, prototype learning, and related areas, to name a few.

**Claims And Evidence:**

Yes

**Claims Explanation:**

This paper presents a new framework, AngularPU, designed to address Positive-Unlabelled learning without relying on explicit negative supervision. By mapping data to the unit hypersphere, the method represents the positive class as a learnable prototype vector. This formulation simplifies the classification task to thresholding the cosine similarity between an embedding and this prototype. It also improves upon current methods, such as negative-risk estimation or pseudo-labelling, which typically depend on strong distributional assumptions or suffer from collapse in high-dimensional settings.

To address the natural tendency of unlabelled embeddings to cluster around the positive prototype, the authors introduce a specialised angular regulariser. This mechanism encourages the dispersion of the unlabelled set across the hypersphere, thereby enhancing class separation.

The paper also provides a strong theoretical foundation, offering proofs for the Bayes-optimality of the angular decision rule and the consistency of the learned prototype. The authors formally demonstrate that under specific directional models (vMF-Uniform), the Bayes decision reduces to thresholding the inner product.

All three reviewers have been very positive, noting that the paper is novel, clear, sound, and theoretically grounded. A few questions they raised were well addressed by the authors, both in their responses and in a revised paper.